# The Mediator kinase module enhances polymerase activity to regulate transcriptional memory after heat stress in Arabidopsis

Tim Crawford [ID], Lara Siebler, Aleksandra Sulkowska, Bryan Nowack, Li Jiang [ID], Yufeng Pan, Jörn Lämke, Christian Kappel [ID] & Isabel Bäurle [ID] ✉

## Abstract

**Plants are often exposed to recurring adverse environmental conditions in the wild. Acclimation to high temperatures entails transcriptional responses, which prime plants to better withstand subsequent stress events. Heat stress (HS)-induced transcriptional memory results in more efficient re-induction of transcription upon recurrence of heat stress. Here, we identified CDK8 and MED12, two subunits of the kinase module of the transcription co-regulator complex, Mediator, as promoters of heat stress memory and associated histone modifications in Arabidopsis. CDK8 is recruited to heat-stress memory genes by HEAT SHOCK TRANSCRIPTION FACTOR A2 (HSFA2). Like HSFA2, CDK8 is largely dispensable for the initial gene induction upon HS, and its function in transcriptional memory is thus independent of primary gene activation. In addition to the promoter and transcriptional start region of target genes, CDK8 also binds their 3′-region, where it may promote elongation, termination, or rapid re-initiation of RNA polymerase II (Pol II) complexes during transcriptional memory bursts. Our work presents a complex role for the Mediator kinase module during transcriptional memory in multicellular eukaryotes, through interactions with transcription factors, chromatin modifications, and promotion of Pol II efficiency.**

**Keywords** Heat Stress; Transcriptional Memory; CDK8; Mediator Kinase Module; RNA Polymerase II
**Subject Categories** Chromatin, Transcription & Genomics; Plant Biology

## Introduction

Exposure to a moderate stress primes plants to better withstand recurrent stress events (Conrath, 2011; Hilker et al, 2016; Lämke and Bäurle, 2017). Boosting the primability of crop plants against abiotic and biotic stresses will enable us to secure their growth and yield under climatic scenarios with more frequent extreme weather conditions (Bita and Gerats, 2013). Biotic and abiotic stressors,

including pathogen attack, drought, salt stress, and heat stress (HS) prime gene expression and thus modify transcriptional responses after recurrent stress exposure (Jaskiewicz et al, 2011; Ding et al, 2012; Kim et al, 2012; Singh et al, 2014; Conrath et al, 2015; Feng et al, 2016; Lämke and Bäurle, 2017; Oberkofler et al, 2021). Such modified responses entail the sustained induction of genes, which persists for several days after the priming stress (known as type I transcriptional memory), and the enhanced re-activation of gene expression after a recurrent HS (type II transcriptional memory) (Lämke et al, 2016; Liu et al, 2018). A similar priming of transcriptional responses occurs in sugar metabolism genes of yeast when it is challenged with an alternative carbon source (D'Urso and Brickner, 2014; D'Urso et al, 2016; Ng et al, 2003) and in the mammalian interferon-γ response (Light et al, 2013). In all of these examples, the transcriptional memory is tightly correlated with hyper-methylation of histone H3 lysine 4 (H3K4me2/3) (Ng et al, 2003; D'Urso et al, 2016; Ding et al, 2012; Jaskiewicz et al, 2011; Feng et al, 2016). In yeast and mammalian cells, H3K4me3 marks recent passage of Pol II and may modify gene loci for rapid re-activation (Ng et al, 2003; D'Urso et al, 2016).

HS memory in *Arabidopsis thaliana* is a model case for transcriptional memory in plants (Oberkofler et al, 2021). Genes that display type I and type II transcriptional memory after HS were identified and a number of components that regulate transcriptional memory were isolated (Stief et al, 2014; Brzezinka et al, 2016; Lämke et al, 2016; Liu et al, 2018; Friedrich et al, 2021). However, the underlying molecular mechanisms, especially those of type II memory, are not understood.

The HEAT SHOCK TRANSCRIPTION FACTORS (HSF) HSFA2 and HSFA3 are required for both types of HS-induced transcriptional memory (Charng et al, 2007; Lämke et al, 2016; Friedrich et al, 2021). In general, members of the HSF family promote transcriptional responses to high temperatures across eukaryotes, in addition to tumorigenesis and aging responses (Li et al, 2017; Gomez-Pastor et al, 2018). Eight of the 21 HSF proteins in *A. thaliana* have been implicated in the HS response (Scharf et al, 2012; Yeh et al, 2012; Ohama et al, 2017). The early HS response is orchestrated by the constitutively expressed *HSFA1* genes (Scharf et al, 2012; Ikeda et al, 2011; Nishizawa-Yokoi et al, 2011; Liu et al, 2011; Yoshida et al, 2011; Yeh et al, 2012). HSFA1 proteins are activated by release from chaperone binding partners

Institute for Biochemistry and Biology, University of Potsdam, Potsdam, Germany. ✉E-mail: isabel.baeurle@uni-potsdam.de

when the chaperones are diverted to other proteins with HS-induced folding problems. Once activated, HSFA1 proteins induce transcription of HS response genes, including chaperones and other HSF genes, until they are eventually inhibited by available chaperones (titration model). The HS-induced HSFA2 and HSFA3 proteins are specifically required for HS memory, and are not involved in the immediate early HS responses. They directly activate genes that show transcriptional memory, such as *APX2* and *HSP22*, and also mediate sustained histone H3K4 hyper-methylation of these genes (Lämke et al, 2016; Friedrich et al, 2021; Kappel et al, 2023). Decreasing H3K4me3 at the *APX2* locus by epigenetic editing decreases HS memory, corroborating a causative role for this chromatin modification in HS memory (Oberkofler and Bäurle, 2022).

Upon DNA binding transcription factors (TFs) activate transcription by promoting the formation of the pre-initiation complex (PIC), the opening of DNA and eventually transcriptional initiation by RNA Polymerase II (Pol II) (Roeder, 2005; Cramer, 2019). Mediator is one of several co-activator complexes that assist this process by regulating the assembly of the PIC and coordinating almost every aspect of transcription (Bourbon, 2008; Kornberg, 2005; Ohama et al, 2021). The highly conserved Mediator complex consists of at least 21 protein subunits, organized into four modules: a core complex (cMed), with head, middle and tail modules (where the tail generally makes contacts with the DNA-bound TFs and the head and middle make contact with Pol II) and a reversibly dissociable cyclin kinase module (CKM) (Abdella et al, 2021; Rengachari et al, 2021; Aibara et al, 2021; Chen et al, 2022b; Tsai et al, 2013, 2014; Nozawa et al, 2017; Robinson et al, 2016; Chen et al, 2021; Li et al, 2021). The role of the CKM is less well understood; it appears to act as an interface between cell signalling and transcription and functions as both an activator and a repressor (Luyties and Taatjes, 2022; Fant and Taatjes, 2019). The CKM consists of the CYCLIN-DEPENDENT KINASE 8 (CDK8), MEDIATOR (MED)12, MED13 and Cyclin C proteins. The primary function of the CKM may be to regulate the interaction of Pol II with cMed during transcription: by sterically blocking the association of cMed with Pol II, it acts as a repressor, but CDK8 can also phosphorylate itself and target proteins to position Pol II for transcription activation (Tsai et al, 2013; Chen et al, 2021; Osman et al, 2021; Rengachari et al, 2021).

Biochemical analyses suggest a conserved subunit composition and structure for plant Mediator (Bäckström et al, 2007; Guo et al, 2020; Mathur et al, 2011; Maji et al, 2019). Plant Mediator subunits are involved in various stress and developmental signalling pathways (Cevik et al, 2012; An et al, 2017; Wang et al, 2019; Zhang and Guo, 2020; Chen et al, 2022a). The plant CKM seems to function as a positive regulator of gene expression in response to environmental stimuli, including drought, cold and biotic stress (Ng et al, 2013; Chen et al, 2019; Crawford et al, 2020; Zhu et al, 2020). The cMed subunits MED14 and MED17 have been implicated in the activation of HS-inducible genes (Ohama et al, 2021).

Here, we identified two subunits of the CKM that are required for the enhanced re-induction of HS memory genes after recurrent HS and for physiological HS memory. HSFA2 recruits CDK8 to common target loci, and both proteins target overlapping sets of memory genes. At these genes, CDK8 promotes H3K4 hyper-methylation and interactions with cMed. We show that the kinase activity of CDK8 is required for regulation of HS memory genes. After recurrent HS, CDK8 binds not only to the promoter, but also the gene body of memory genes, where it affects Pol II dynamics. Our findings indicate that CDK8 is involved in regulating Pol II activity by resolving blocked Pol II complexes and/or by efficiently recycling it for additional transcription cycles.

# Results

## *REINDUCTION* (*REIN*) genes are required for enhanced re-induction of *APX2* after recurring HS

To identify regulators of HS-induced transcriptional memory we performed a forward mutagenesis screen for modifiers of *pAPX2::LUCIFERASE (LUC)* hyper-induction after recurrent HS, where a 592 bp promoter fragment of *APX2* confers HS-induced transcriptional memory onto the *LUC* reporter (Liu et al, 2018). We recovered three mutants with reduced LUC activity after recurrent HS (priming and triggering, $P + T$) relative to the parent (Fig. 1A,B), but normal or largely normal activity after a single HS (P or T); we called these mutants *reinduction* (*rein*) *1-1*, *rein1-2*, and *rein2*. In line with this finding, all three *rein* mutants displayed defective HS memory at the physiological level (Fig. 2A,B, see below), indicating that HS-induced transcriptional memory is required for physiological HS memory.

## *REIN1* and *REIN2* encode components of the CKM

To identify the genetic lesions underlying the re-induction phenotype in the *rein* mutants, we combined recombination breakpoint mapping and genome re-sequencing. In *rein1-1*, we found a nonsense mutation in exon 2 of the *CDK8* gene (*At5g63610*) that introduced a premature stop codon in place of Gly425 (Fig. 1C). In *rein1-2*, a nonsense mutation in the same exon introduced a stop codon in place of Trp200 (Fig. 1C), indicating that *rein1-1* and *rein1-2* are allelic. In *rein2* we detected a missense mutation in exon 3 of *MED12* (*At4g00450*, Fig. 1C), introducing a missense mutation (Gly88Glu) in the N-terminal domain that in human cells is required for regulating CDK8 activity (Knuesel et al, 2009b; Klatt et al, 2020; Luyties and Taatjes, 2022). To confirm that *rein1-1* and *rein1-2* are allelic to *cdk8*, we crossed them with a characterized *cdk8* T-DNA line (Crawford et al, 2020) and analysed *pAPX2::LUC* re-induction in the progeny. The F1 progeny of *rein1-1* x *cdk8* and *rein1-2* x *cdk8*, but not *rein1* x Col-0, showed strongly reduced re-induction of *pAPX2::LUC* after $P + T$ (Fig. 1D, Appendix Fig. S1A). A corresponding result was obtained after crossing *rein2* to the *cct-1* mutant allele of *MED12* (Appendix Fig. S1B; Gillmor et al, 2010), and after crossing the *rein1-1* and *rein1-2* lines (Appendix Fig. S1C), respectively. In addition, a genomic construct expressing *pCDK8::CDK8-GSYellow* (*GSY*) complemented the physiological HS memory phenotype of *rein1-1* (Fig. 2B). Thus, loss of CKM function causes the HS memory defects in *rein1* and *rein2*.

## *CDK8* and *MED12* are required for physiological HS memory

We next assessed the physiological HS memory of *rein1, rein2, cdk8* and *med12* by treating seedlings with a strong HS 3 d after an

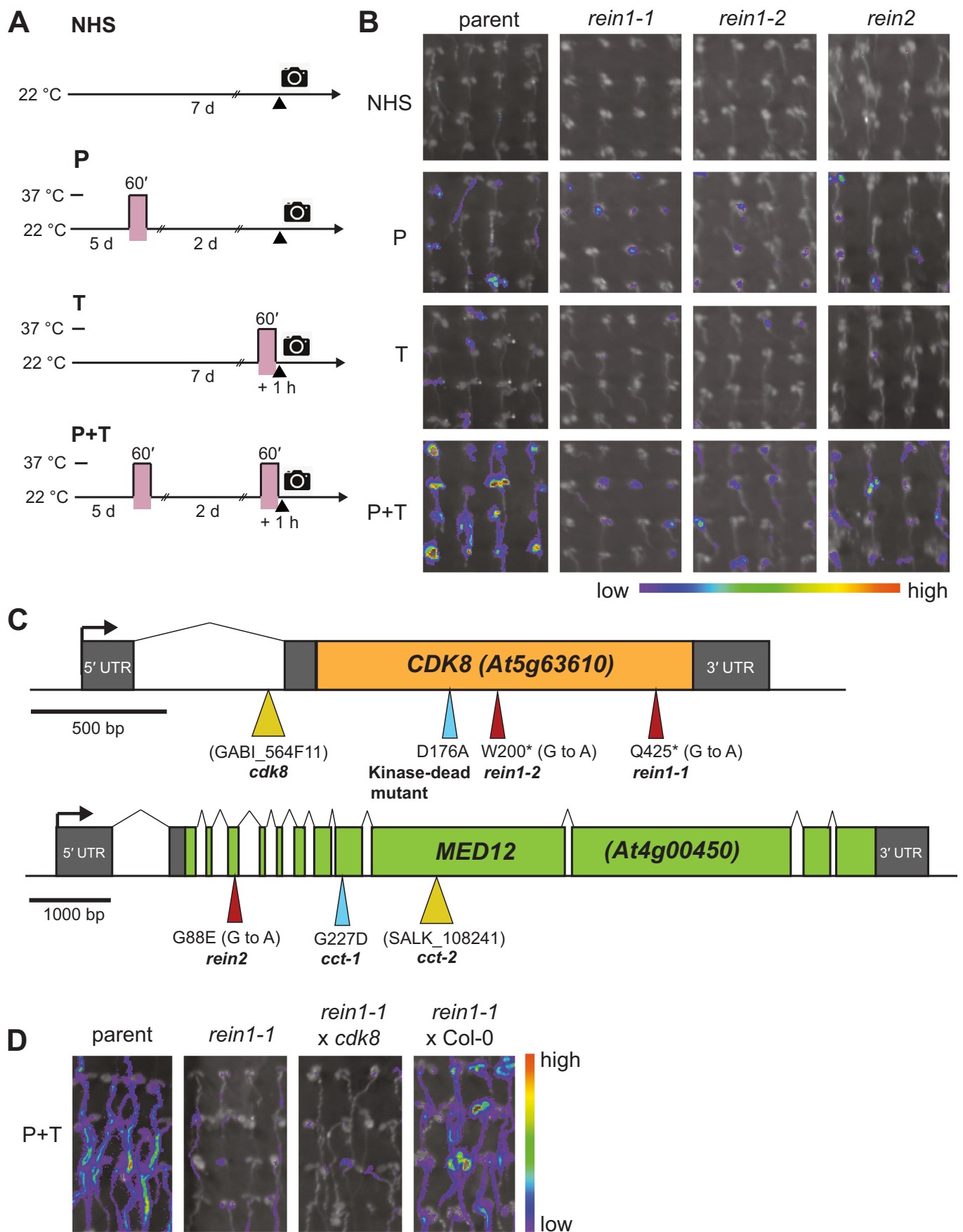

**Figure 1. CKM mutants are required for HS-induced transcriptional memory.**

(A) Treatment scheme for LUC-based HS memory assay. 5 d-old seedlings were subjected to the indicated treatments (no HS (NHS) or a single HS on d 5 (P) or d 7 (T), or recurrent HS on d 5 and d 7 (P + T)). LUC activity from the *pAPX2::LUC* reporter gene was monitored on d 7 with 1 h recovery after T. (B) Bioluminescence signal of seedlings of *rein1-1*, *rein1-2*, *rein2* mutants and their parental line after the indicated treatments. The signal intensity was converted to the indicated false colour scale. (C) Schematic representations of the *CDK8* and *MED12* loci. The positions of EMS-induced mutations, T-DNA insertion sites and induced mutations are indicated. (D) Allelism of *rein1-1* and *cdk8*. Bioluminescence in seedlings after repeated HS (P + T) was not complemented in the F1 progeny of *rein1-1* crossed with *cdk8*, but was complemented after crossing to Col-0. Source data are available online for this figure.

acclimating HS (ACC) (maintenance of acquired thermotolerance (maTT) assay, cf. Methods, Fig. 2A; Charng et al, 2007; Stief et al, 2014). The *rein1-1* to *rein2*, *cdk8* and *cct-2* mutants displayed significantly reduced growth and survival relative to the parent, but slightly less severe than *hsfa2* (Fig. 2B–H). To test whether the defect in thermotolerance was specific for the memory phase, we also tested their acquisition of thermotolerance (aTT), i.e. the capacity to withstand a strong HS immediately after ACC, as well as their basal thermotolerance (bTT), i.e. the capacity to withstand a strong HS without prior acclimation. *Rein1-1* and *rein1-2* displayed slightly reduced aTT (Fig. EV1A–D). BTT was not affected in *rein1-1*, *rein2*, and only slightly affected in *rein1-2* (Fig. EV2A–D). The *cdk8* T-DNA allele displayed similar phenotypes as *rein1-1* and *rein1-2* in all three assays (Figs. 2D–G, EV1B–D, and EV2B–D). Together, these findings indicate that *CDK8* and *MED12* contribute to physiological HS memory, in addition to a minor role in the acquisition of thermotolerance.

## CDK8 and MED12 are required for expression of type I and II HS memory genes

We next characterized the transcriptional defects underlying the impaired HS memory in *cdk8/rein1* and *med12/rein2*. To assess type II transcriptional memory (enhanced re-induction in response to recurrent HS), we subjected seedlings of *rein1-1*, *rein1-2*, *rein2*, and *CDK8-GSY rein1-1*, to a type II HS regime (Fig. 3A) and analysed gene expression by quantitative (q)RT-PCR. Hyper-induction of the described type II memory genes *APX2*, *pAPX2::LUC*, *MIPS2*, *LPAT5*, *XTR6* and *LACS9* (Liu et al, 2018) was significantly reduced in *rein1-1* and *rein1-2* relative to the parent, and was rescued in *CDK8-GSY rein1-1* (P + T relative to T; Fig. 3B). In *rein2*, enhanced re-induction was also significantly impaired for these genes. In contrast, the induction of HS-induced non-memory genes *HSP70* and *HSP101* was not affected, with the exception of a significant reduction of *HSP70* in *rein1-1* after T. These results were confirmed in a time course expression analysis where we sampled every 15 min after the onset of T for 60 min, in primed or naïve plants (Appendix Fig. S2). We also analysed the sustained induction of type I memory genes over 3 d following an ACC treatment. Starting from 28 h after ACC, the expression of *APX2*, *HSP18.2*, and *HSP21* was significantly reduced in *rein1-1* and *rein1-2* relative to the parent (Fig. EV3A,B). In *rein2*, expression was not affected during the memory phase (Fig. EV3A,B); instead, the initial induction of *APX2*, *HSP18.2*, *HSP21* and the non-memory HS-responsive gene *HSP70* were significantly reduced. These results indicate that *CDK8* is required for type I and type II transcriptional memory, while *MED12* is required for type II transcriptional memory. Thus, the CKM promotes HS-induced transcription and its memory.

## CDK8 and HSFA2 regulate overlapping sets of type II memory genes

To globally assess the contribution of *CDK8* to type II transcriptional memory, we performed RNA-sequencing on *cdk8* and *hsfa2* mutants after a type II HS regime (Fig. 4A). Analysis of differential gene expression in Col-0 identified 152 genes displaying +/++ behaviour (up-regulated in T/NHS, P + T/P and P + T/T conditions) and 185 displaying 0/+ behaviour (up-regulated in P + T/P and P + T/T only; Fig. 4B, Dataset EV1). Both gene sets were highly overlapping with a previous microarray-based study (Liu et al, 2018) and included known candidates such as *APX2*, *MIPS2*, *XTR6* (+/++) and *LPAT5* and *LACS9* (0/+), respectively. Neither set was significantly enriched for any gene ontology or KEGG pathway term. Both sets showed reduced induction in *hsfa2* and *cdk8* after P + T, but not after any of the other treatments (Fig. 4C,D, Dataset EV2). In comparison, early HS-inducible genes were not affected (in *hsfa2*), or only slightly affected (*cdk8*) (Fig. 4C,D, Dataset EV2). Consistently, a large number of +/++ genes showed a reduced hyper-induction in either mutant compared to Col-0 (Fig. 4E). We also analysed the total number of memory genes that lost hyper-induction after P + T in the mutants (Fig. 4F). Of the 152 genes in the +/++ group in Col-0, 116 (76%) and 92 (61%) genes were non-responsive in *hsfa2* and *cdk8*, respectively (Fig. 4F); of the 185 genes in the 0/+ group, 165 (89%) and 157 (86%) were not up-regulated in *hsfa2* and *cdk8*, respectively (Fig. 4F). The non-responsive genes were strongly overlapping between *hsfa2* and *cdk8* (53% of +/++ genes and 76% of 0/+ genes). In summary, a large proportion of memory genes in either class require *CDK8* for their expression, and these genes overlap strongly with the *HSFA2*-dependent genes. Thus, *CDK8* and *HSFA2* activate overlapping sets of genes during HS-induced transcriptional memory.

## CDK8 binds HS memory genes

As CDK8 is required for the expression of the type II HS memory genes *APX2* and *MIPS2*, we wondered whether it physically associates with these genes. Thus, we performed chromatin immunoprecipitation-qPCR (ChIP-qPCR) on a complementing *35S::CDK8-MYC* line (Zhu et al, 2014) after type II HS treatment (Fig. 4A). At both loci we observed a small enrichment of CDK8 at the promoter before HS (NHS; Fig. 5). Similarly, MED12 was enriched at *MIPS2* in a previous study (Liu et al, 2020). After T, CDK8-MYC occupancy remained at baseline levels for *APX2* and was slightly elevated for *MIPS2*. In contrast, at P + T, CDK8 binding increased sharply at both loci throughout the gene body and especially towards their 3'-ends. In comparison, at the HS-induced non-memory gene *HSP70*, CDK8 was strongly and

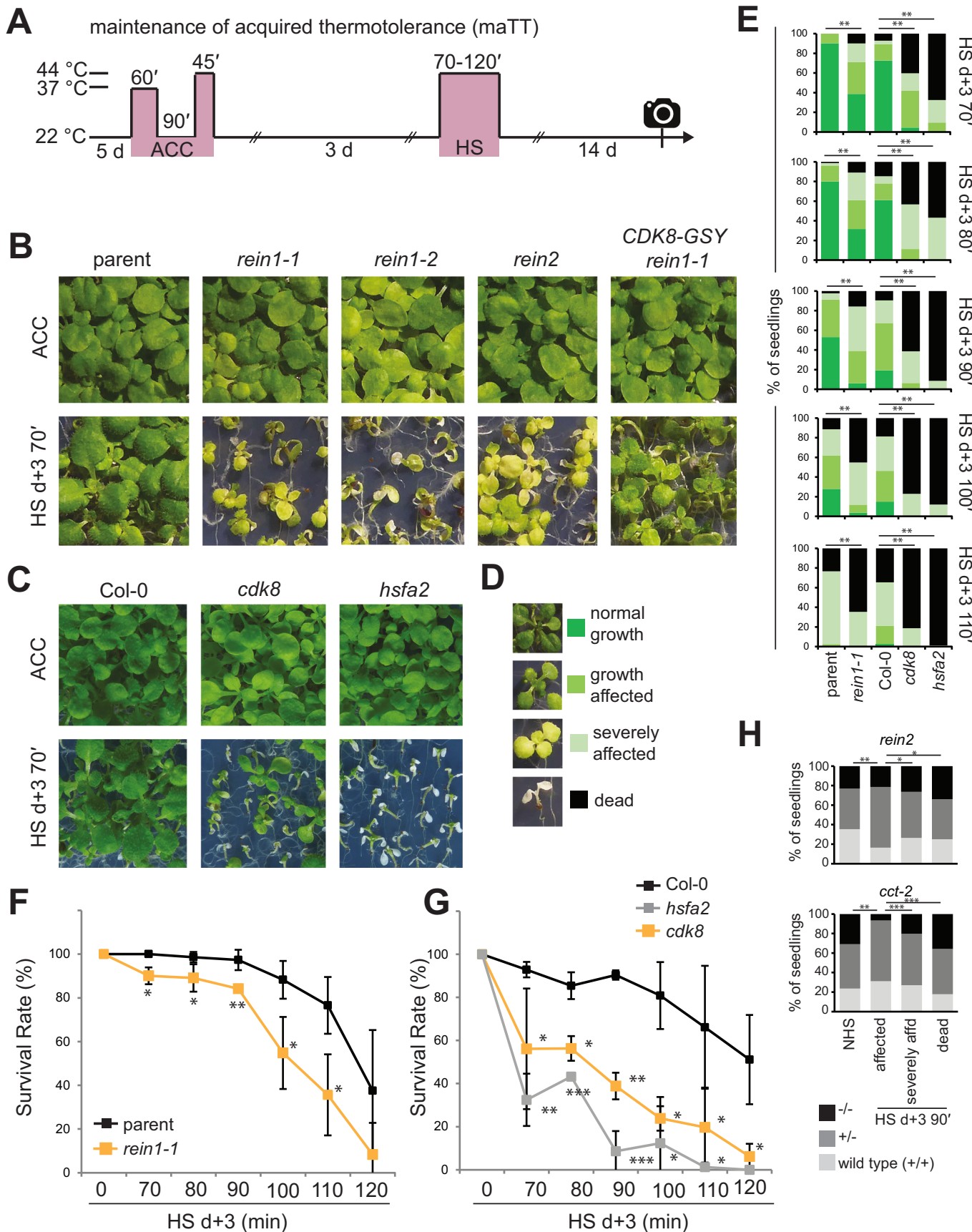

◀ **Figure 2.   The CKM is required for physiological HS memory.**

(A) Treatment scheme for HS memory (maintenance of acquired thermotolerance) assay. Seedlings were exposed to a HS (44 °C for 70-120 min) 3 d after ACC (which was applied 5 d after germination). Survival and phenotypic category were analysed after 14 d of recovery (camera symbol). Three or four biological replicate experiments were performed. (B, C) Representative HS memory assay with the *rein1-1*, *rein1-2* and *rein2* mutants, the complementing *CDK8-GSY rein1-1* line and the parent (B), as well as the *cdk8* and *hsfa2* mutants and Col-0 wild type (C). (D) Examples of phenotypic categories used in (E, H). (E) Distribution of phenotypic categories for *rein1-1*, *cdk8*, *hsfa2*. Asterisks indicate significant differences to their respective controls. **$p < 0.005$; Fisher's exact test. (F, G) Survival rates for *rein1-1*, *cdk8*, *hsfa2*. Error bars indicate the mean ± SEM of three independent biological replicate experiments. Asterisks indicate significant differences to their respective controls. *$p < 0.05$; **$p < 0.005$; ***$p < 0.0005$, unpaired, two-tailed *t* test. (H) Co-segregation of *rein2* and *cct-2* alleles and reduced HS memory, measured at HS d + 3 90′. Individual seedlings from a family segregating for the indicated mutant were phenotyped and genotyped. Higher damage classes are enriched for homozygous mutants, indicating a defective HS memory in *rein2* and *cct-2*. *$p < 0.05$; **$p < 0.01$, ***$p < 0.001$, determined by $X^2$ test. Source data are available online for this figure.

equally enriched at T and P + T (Fig. 5). Notably, total CDK8 protein levels were largely constant during the treatments, ruling out that altered protein levels drove the observed dynamics in enrichment (Fig. 6C). Thus, CDK8 is enriched throughout promoters and gene bodies of HS memory genes in P + T, suggesting a direct role in their enhanced re-induction after recurrent HS.

## CDK8 interacts with HSFA2

The Mediator co-activator complex is recruited by gene-specific transcription factors (Zhu et al, 2020). HSFA2 binds after HS to the promoters of HS memory genes in a region overlapping with CDK8 binding (Fig. 5; Kappel et al, 2023). As CDK8 and HSFA2 are both required for HS-induced transcriptional memory, we hypothesized that they interact directly. We tested this notion using the split-luciferase complementation assay in transiently transformed *Nicotiana benthamiana* leaves. Co-expression of N-RLuc-HSFA2 or N-RLuc-HSFA3 with the C-RLuc-CDK8 fusion protein resulted in reconstituted RLUC activity, whereas co-expression with the corresponding N/C-RLUC fragments did not (Fig. 6A). In addition, FLAG-HSFA2 or FLAG-HSFA3 were immunoprecipitated by HALO-CDK8 but not HALO alone after co-expression in wheat germ extracts (Fig. 6B). To probe the CDK8-HSFA2 interaction in stable transgenic *A. thaliana*, we crossed *35S::CDK8-MYC* and *pHSFA2::FLAG-HSFA2* (Friedrich et al, 2021) and obtained doubly homozygous lines and sampled at various time points after the end of an ACC treatment (Fig. 6C,D). Consistent with previous reports, FLAG-HSFA2 was induced during HS. The constantly expressed CDK8-MYC co-immunoprecipitated with FLAG-HSFA2 between 0 and 4 h after ACC, peaking immediately after ACC (Fig. 6C). At 28 h after ACC, CDK8-MYC no longer co-immunoprecipitated with FLAG-HSFA2, though both proteins were still present. Thus, CDK8 and HSFA2 interact in vivo, preferentially during the early recovery phase.

## HSFA2 recruits CDK8 to memory genes

Considering the interaction of CDK8 and HSFA2 and the overlapping binding in the promoters of target genes, we wondered if HSFA2 plays a role in recruiting CDK8 to HS memory genes. We thus introduced *35S::CDK8-MYC* into the *hsfa2* background and performed ChIP-qPCR. The recruitment of CDK8-MYC to *APX2* and *MIPS2* was almost completely abolished in the absence of *HSFA2* (Fig. 5). In contrast, CDK8-MYC binding to the non-memory gene *HSP70* was only slightly reduced. Together, these findings indicate that HSFA2 interacts with the CKM

through CDK8 and recruits it to HS memory genes after recurrent HS.

## The cMed subunit MED23 shows overlapping binding at gene bodies with CDK8

The CKM dynamically associates with cMed, and thus the occupancy of CKM and cMed may vary at target genes (Kagey et al, 2010; Zhu et al, 2006; Jeronimo et al, 2016). We therefore asked whether cMed is enriched at memory genes after HS together with CDK8 by generating a tagged line of the tail module subunit MED23 (*pMED23::HA-MED23*). ChIP-qPCR showed that MED23 is enriched at *APX2* and *MIPS2* after P + T, and at *MIPS2* to a lesser extent also after T. MED23 occupancy increased towards the 3′ ends, as was observed for CDK8 (Fig. EV4). At *HSP70*, MED23 was enriched similarly in T and P + T. Thus, CDK8 appears to bind HS genes together with cMed.

## The kinase activity of CDK8 is essential for its function in transcriptional memory

*A. thaliana* CDK8 has previously been shown to regulate defence-related gene expression via kinase-dependent and -independent mechanisms (Zhu et al, 2014). To test if the kinase activity of CDK8 was required for HS memory, we employed a mutated CDK8 in which the kinase activity was inactivated by the Asp176Ala mutation (Fig. 1C, Zhu et al 2014). *35S::CDK8-MYC/cdk8* and *35S::CDK8-kinase dead (KD)-MYC/cdk8*, seedlings were treated with a type II HS regime (Fig. 4A) and transcript levels of *APX2*, *MIPS2* and *HSP70* were determined by qRT-PCR. While the wild-type CDK8-MYC fully complemented the hyper-induction defects of *APX2* and *MIPS2* in *cdk8* (Fig. 6E), the kinase-dead protein did not complement the mutant phenotype, indicating that the kinase activity of CDK8 is required for HS-induced transcriptional memory.

## CDK8 is required for H3K4me3 accumulation at the *APX2* memory locus

Elevated levels of H3K4me3 are tightly associated with HS-induced transcriptional memory and depend on HSFA2 (Lämke et al, 2016; Oberkofler and Bäurle, 2022; Liu et al, 2018). (Yamaguchi et al, 2021). We next asked whether CDK8 is required for H3K4 hyper-methylation at HS memory genes. H3K4me3 was enriched downstream of the transcriptional start site of *APX2*, at 2 d after HS (P) and immediately after a single HS (T) relative to NHS (Fig. 7). The highest enrichment of H3K4me3, however, was observed after recurrent HS (P + T), consistent with previous observations (Liu et al, 2018; Oberkofler and Bäurle, 2022). In the *cdk8* mutant, the enrichment of

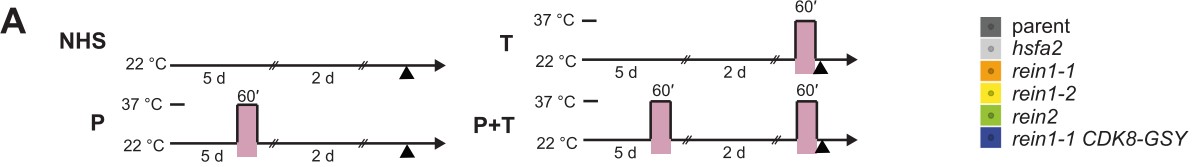

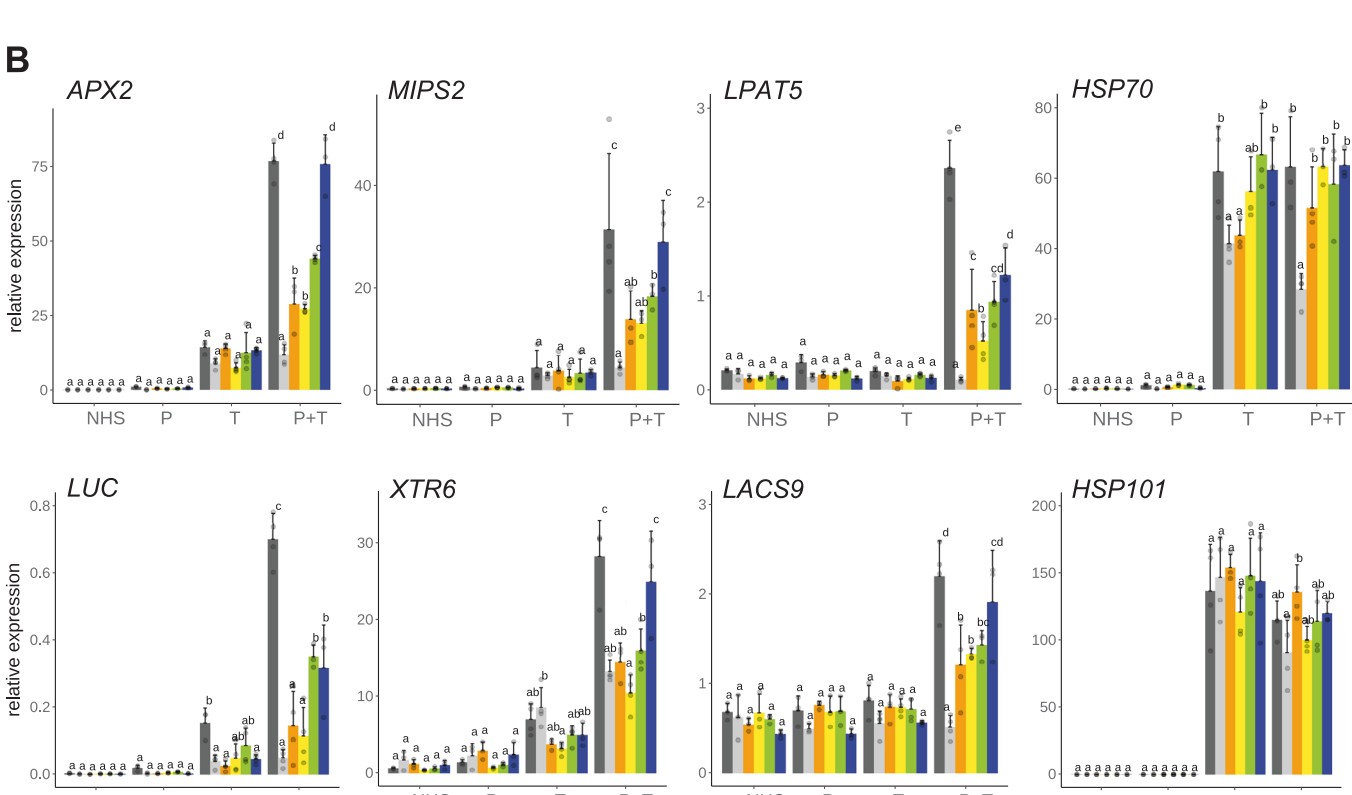

**Figure 3. Type II transcriptional HS memory gene expression is impaired in *rein* mutants.**

(A) Treatment schema for the type II HS memory assay. NHS, no heat stress; P, HS on d 5 only (primed); T, HS on d 7 only (triggered); P + T, HS on both d 5 and d 7 (primed + triggered). Each HS consisted of 60 min at 37 °C. Seedlings were sampled at the same time on d 7 (arrowhead). (B) Seedlings were treated as indicated and relative transcript levels of six type II memory genes (*APX2, pAPX2::LUC, MIPS2, LPAT5, XTR6, LACS9*) and two HS-induced non-memory genes (*HSP70, HSP101*) were measured by qRT-PCR and normalized to the expression of *At4g26410*. Data are mean ± SEM of three to four independent biological replicate experiments, along with individual data points. Transcript levels were statistically evaluated for all genotypes within each timepoint by ANOVA followed by Tukey's HSD test (*p* < 0.05). Genotypes are assigned one or more letters based on their statistical group. Genotypes sharing one letter are not significantly different. Source data are available online for this figure.

H3K4me3 was reduced after P and P + T, suggesting that CDK8 promotes H3K4 trimethylation at *APX2*. At *MIPS2, HSP70* and *HSP101*, H3K4me3 was already enriched in NHS compared with negative genomic controls, suggesting divergent mechanisms at these loci. At *MIPS2*, we observed a small increase in H3K4me3 at P + T in the wild type, which was reduced in *cdk8*. At the non-memory loci *HSP70* and *HSP101*, the enrichment at NHS did not increase further after any of the treatments and was independent of CDK8. We also analysed H3K9ac, which closely correlates with transcriptional activity, in particular PIC assembly and transcription initiation (Lämke et al, 2016; Leng et al, 2020). H3K9ac was enriched after T and P + T at all genes tested, but not P or NHS, and this enrichment was independent of CDK8 (Fig. 7). Thus, the CDK8-dependent dynamics of H3K4 hyper-methylation are specific and not merely a consequence of acute transcriptional activity.

## CDK8 coordinates RNA Pol II during HS-induced transcriptional memory

The CKM regulates the activity of Pol II (Fant and Taatjes, 2019; Osman et al, 2021). We thus investigated the dynamics of the Pol II major subunit, NRPB1, at the CDK8-dependent genes *APX2* and *MIPS2* after a type II HS regime in wild type and *cdk8* (Fig. 8A–C). Overall Pol II occupancy was low at *APX2* and *MIPS2* loci in the NHS and P conditions. After T, Pol II occupancy increased at all genes tested, consistent with their transcriptional activation. After P + T, Pol II occupancy increased further at *APX2* and *MIPS2*, especially towards the 3′-ends of the genes. Conversely, the non-memory HS genes *HSP70* and *HSP101* showed decreased Pol II occupancy after P + T relative to T. In *cdk8*, we unexpectedly found increased Pol II enrichment at *APX2* and *MIPS2 after* P + T, and this was most

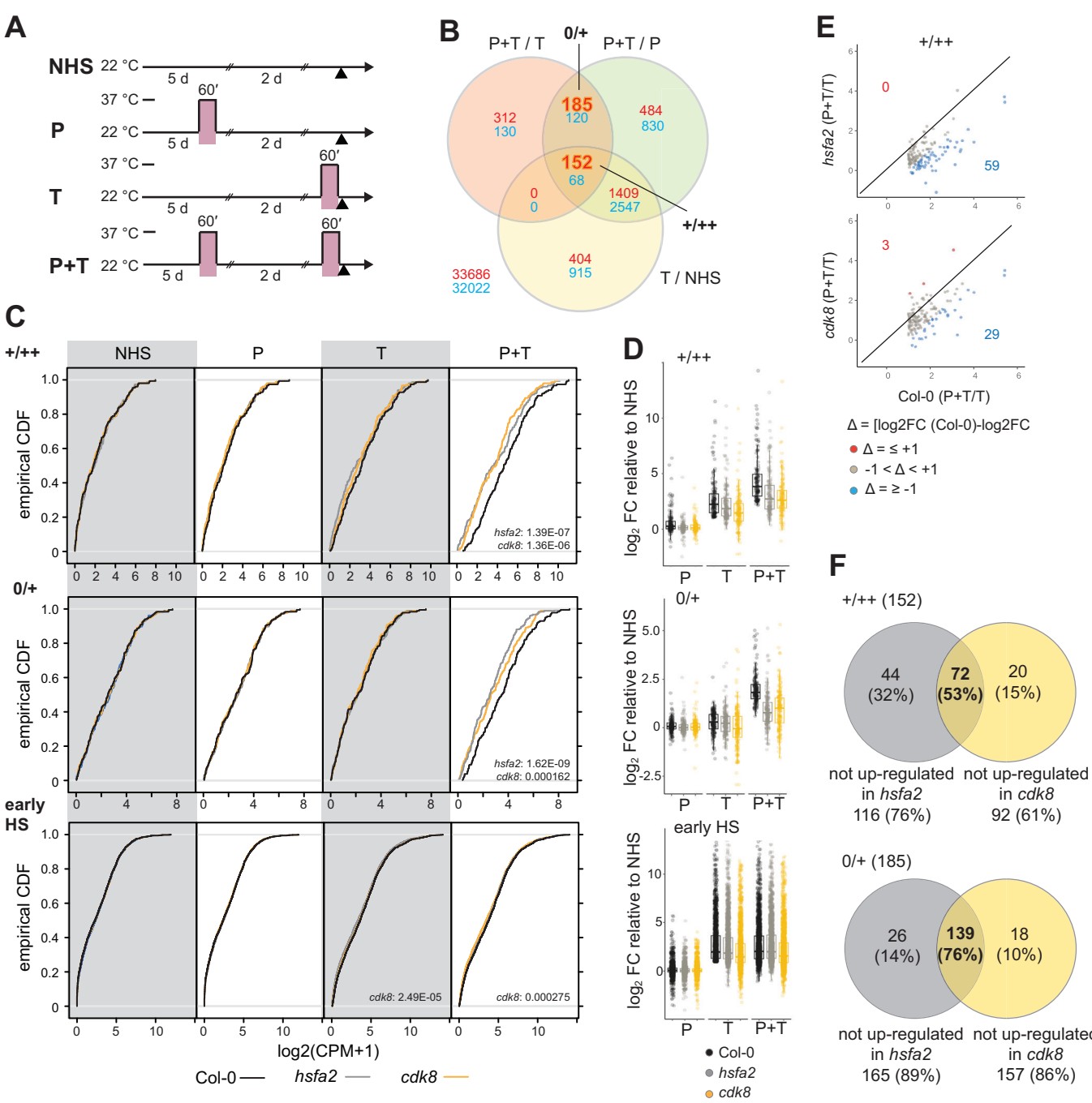

**Figure 4. CDK8 and HSFA2 regulate an overlapping set of type II transcriptional HS memory genes.**

(A) Sampling regime for the RNA-sequencing experiment. NHS, no heat stress; P, HS on d 5 only (primed); T, HS on d 7 only (triggered); P + T, HS on both d 5 and d 7 (primed + triggered). Each HS consisted of 60 min at 37 °C. Seedlings were sampled at the same time on d 7 (arrowhead). (B) Identification of genes with type II transcriptional HS memory in the Col-0 wild type (+/++ and 0/+ subtypes). The numbers of up-regulated genes from each category are shown in red, and down-regulated genes in blue; memory gene categories are highlighted. (C) empirical cumulative distribution function plots of expression for all genes in the +/++ (top) or 0/+ memory gene (middle) or early HS gene groups (bottom) at the indicated timepoints and genotypes. Numbers on panels indicate significantly different distributions relative to Col-0 ($p < 0.05$, KS test). (D) Log$_2$ fold-change at P, T and P + T (relative to NHS) of each transcript in the +/++, 0/+ memory gene or early HS gene groups. Boxes show the mean, upper and lower quartile, and upper and lower whiskers show the largest or smallest value no more than 1.5x the interquartile range from the upper or lower quartile, respectively, for each genotype and treatment ($n = 152$, 185 and 1813 for +/++, 0/+ and early HS groups, respectively). (E) Comparison of hyper-induction (P + T/T) between Col-0 and *hsfa2* or *cdk8* for +/++ genes. Genes in red are more strongly hyper-induced in the mutant, genes in blue are less hyper-induced in the respective mutant. (F) Overlap of non-responsive +/++ (upper panel) and 0/+ memory genes (lower panel) in *hsfa2* and *cdk8* mutants. Venn diagrams indicate the number and percentage of non-responsive genes in each mutant. Source data are available online for this figure.

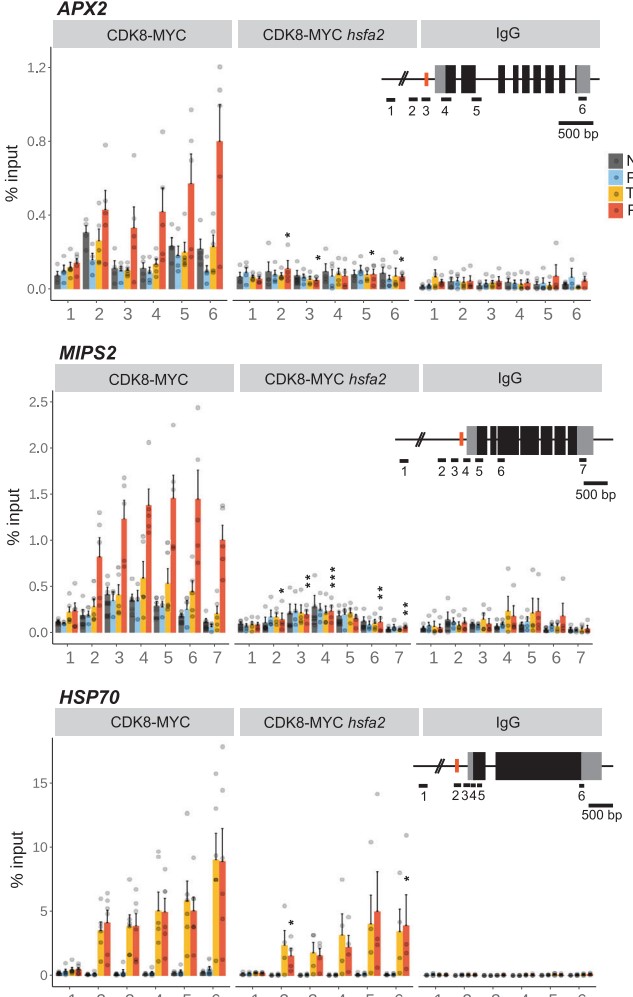

**Figure 5. CDK8 is recruited to HS memory gene loci in an HSFA2-dependent manner.**

CDK8-MYC seedlings in the wild type or *hsfa2* mutant background were subjected to a type II HS regime (cf. Fig. 4A). Occupancy of CDK8-MYC was determined by ChIP-qPCR using antibodies against the MYC tag or IgG as a negative control. Amplicons targeted the sites indicated on the gene models; red bars indicate binding sites for HSFA2 according to (Kappel et al, 2023). Mean ± SEM as well as individual data points from five independent biological replicate experiments are shown. Asterisks denote significant differences in CDK8 occupancy in *hsfa2* relative to the wild type background (*p < 0.05; **p < 0.005; ***p < 0.0005; unpaired, two-tailed *t* test).

pronounced towards their 3′-ends. In addition, after P, Pol II was enriched around the TSS of *APX2* in *cdk8*, but not wild type. The activity of Pol II is regulated by the phosphorylation of its C-terminal domain repeats, namely Serine-5 (Ser5p) and Serine-2 (Ser2p). We found that the distribution of these modifications at the HS gene loci was similar to that of total Pol II (Fig. 8A–C). After P + T, Ser5p- and Ser2p-modified Pol II were both more strongly enriched in *cdk8* than in wild type, despite decreased gene expression; this difference was most pronounced for Ser2p (Fig. 8C).

Together, these data indicate that CDK8 strongly affects the occupancy of Pol II at HS memory gene loci. Surprisingly, Pol II occupancy was higher in *cdk8* despite a lower level of polyadenylated

transcripts (as measured by qRT-PCR and RNA-seq). To estimate the levels of nascent elongating transcripts, we immunoprecipitated NRPB2 complexes after a type II HS regime and quantified the associated transcripts (Kindgren et al, 2020). In wild type this approach detected high levels of nascent elongating transcripts of *APX2* and *MIPS2* after P + T (Fig. EV5A,B); in contrast, transcripts were hardly detected at NHS or P, and only slightly elevated after T. In *cdk8*, the levels of nascent transcripts of *APX2* and *MIPS2* were significantly lower than in the wild type after P + T, consistent with the reduced level of mature transcripts in *cdk8*. For *HSP70* and *HSP101*, Pol II-associated transcripts were highly elevated after T and P + T in wild type (Fig. EV5C). In *cdk8 HSP70 and HSP101* levels were lower after T and unchanged after P + T. To confirm these findings we analysed unspliced transcripts directly after the same treatments. We observed higher levels of *APX2* and *MIPS2* at P + T than at T and this hyper-induction was lost in *cdk8* (Fig. EV5D). For *HSP70* and *HSP101*, nascent transcript levels were less increased in *cdk8* at T and similar to wild type at P + T. Thus, the direct measurement of unspliced transcript levels fully agrees with Pol II-associated transcript levels. Thus, efficient transcriptional elongation at HS memory genes after P + T requires *CDK8*. Together with the increased Pol II binding after P + T, our data suggest that in the absence of CDK8 non-productive Pol II complexes accumulate at the 3′-ends of HS memory genes.

## Discussion

Through a forward genetic screen, we identified two components of the CKM, CDK8 and MED12, as being required for HS-induced transcriptional memory. The mutants are deficient in physiological HS memory and this correlates with loss of enhanced re-induction of transcription after recurrent HS (Figs. 2 and 3). These phenotypes are consistent with an assumed role of the CKM in rapid stimulus-dependent gene induction (Luyties and Taatjes, 2022). *A. thaliana* MED12 has previously been implicated in the transition from low to high expression states in response to environmental changes (Liu et al, 2020). Overall, *rein2/med12* has weaker phenotypes than *rein1/cdk8* (Fig. 3). *Rein2* has a missense mutation in a highly conserved Gly, Gly44 in mammals (Knuesel et al, 2009b; Klatt et al, 2020; Luyties and Taatjes, 2022), which is required for stimulating kinase activity of CDK8, and may thus retain some functionality. Hence, while CDK8 kinase activity may be impaired in *rein2*, kinase-independent functions of the CKM such as allosteric inhibition of the Pol II binding site on cMed (Tsai et al, 2013; Osman et al, 2021) may still be intact. This is in line with our finding that a kinase-dead version of CDK8 is unable to complement the transcriptional memory defect, while it has been reported to partially complement other phenotypes of the *cdk8* mutant (Zhu et al, 2014; Fig. 6E). It is also consistent with the observation that loss-of-function alleles of *MED12* have strong developmental phenotypes that were not observed in *rein2* (Gillmor et al, 2010). Alternatively, CDK8 may have functions that are independent of MED12.

The *cdk8* mutant is impaired in both types of HS-induced transcriptional memory and has strongly reduced H3K4me3 hyper-methylation at *APX2* (Figs. 3 and 7). These defects are highly similar to those found in *hsfa2* mutants for H3K4me3 and H3K4me2 (Lämke et al, 2016). Of note, the anti-H3K4me3 antibody used here may also show non-specific binding to

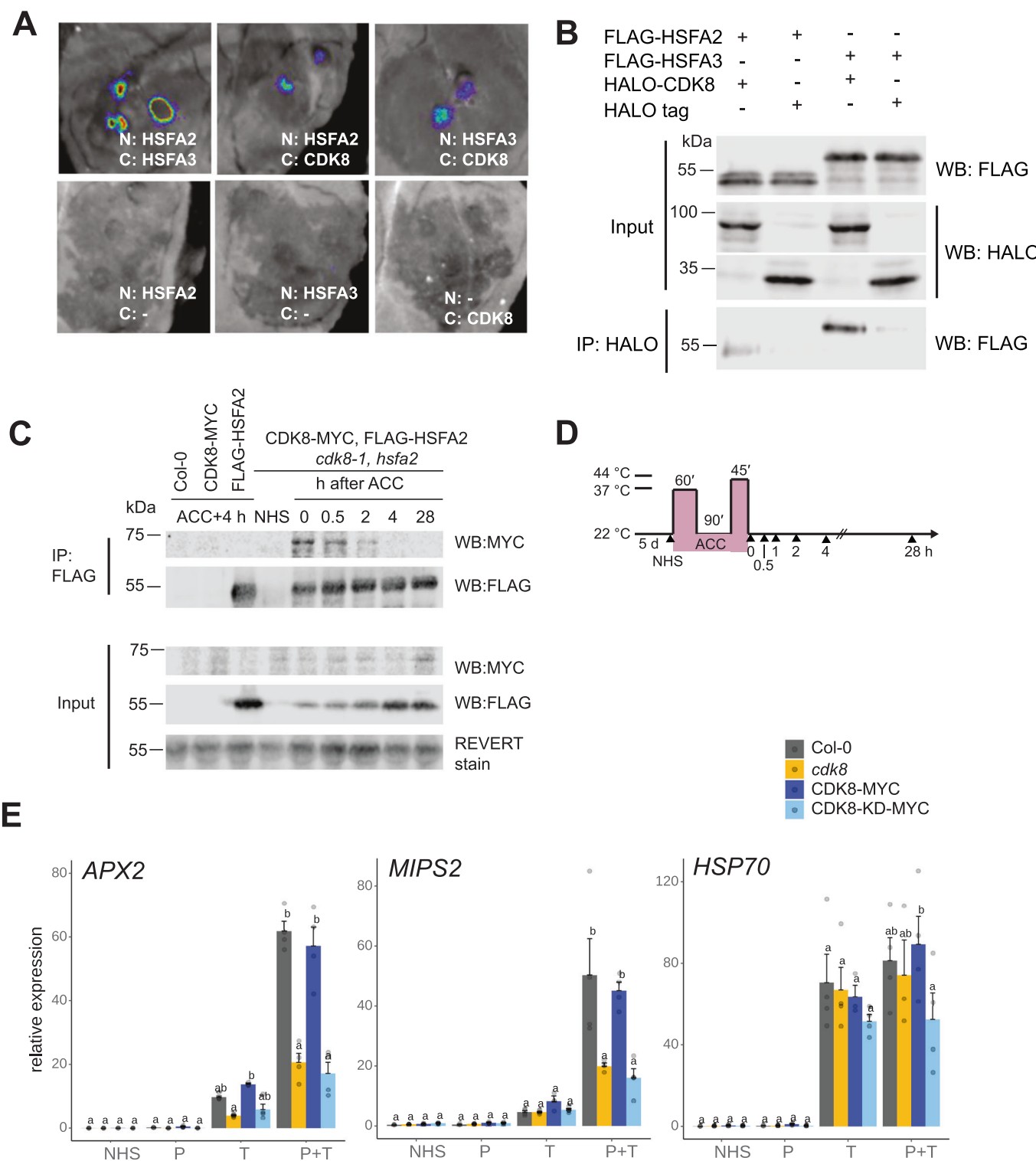

H3K4me2 (Shah et al, 2018). In addition, CDK8 and HSFA2 proteins interact and HSFA2 is required to recruit CDK8 to HS memory loci (Figs. 5 and 6). This is notable, as HSFA2 is dispensable for the initial HS-mediated induction of those genes, which is mediated by the HSFA1 isoforms (Liu et al, 2011). Thus, CDK8 likely discriminates between different HSF proteins to

specifically function in HS memory. This is consistent with previous reports where cMed (MED14/17) was found to be required for the HS response at some but not all HS-responsive genes (Ohama et al, 2021). While binding of cMed to these genes was abrogated in the *hsfa1* quadruple mutant, the authors did not observe a direct interaction with HSFA1s. As HSFA2 induction is

**Figure 6. CDK8 interacts with HSF transcription factors and requires kinase activity during HS memory.**

(**A**) Split-luciferase complementation assay showing the interaction between HSFA2/ HSFA3 and CDK8 in *Nicotiana benthamiana*. Constructs expressing N-RLuc fusions with HSFA2 or HSFA3 and C-RLuc fusions with CDK8 were co-transfected into fully expanded leaves. C-RLuc-HSFA3 was used as a positive control for N-RLuc-HSFA2, and respective empty vectors serve as negative controls. A representative of two biological replicate experiments is shown. (**B**) In vitro co-immunoprecipitation assay showing the interaction between HSFA2/HSFA3 and CDK8. FLAG-tagged HSFA and HALO-tagged CDK8 proteins were co-expressed in wheat germ total protein extract. Proteins were immunoprecipitated using anti-HALO magnetic beads and detected by immunoblotting with antibodies against FLAG and HALO. Empty HALO tag served as a negative control. A representative of two biological replicate experiments is shown. (**C**) In vivo co-immunoprecipitation assay showing the interaction dynamics of CDK8 and HSFA2 during HS memory. CDK8-MYC FLAG-HSFA2 expressing seedlings, as well as the respective control lines and the Col-0 wild type, were grown for 5 d before being subjected to a type I HS regime and sampled at the timepoints indicated in (**D**). Proteins were immunoprecipitated from total protein extracts using anti-FLAG beads and detected by immunoblotting with antibodies against MYC or FLAG, respectively. REVERT total protein stain of the membrane is shown as loading control. (**E**) Kinase activity of CDK8 is required for type II transcriptional memory. CDK8-KD-MYC (carrying a D176A mutation) and CDK8-MYC seedlings in *cdk8* background were subjected to a type II HS regime (cf. Fig. 4A). NHS, no heat stress; P, HS on d 5 only (primed); T, HS on d 7 only (triggered); P + T, HS on both d 5 and d 7 (primed + triggered); each HS consisted of 37 °C for 60 min, and seedlings were sampled on d 7. Data shown are the mean ± SEM of transcript levels of *APX2*, *MIPS2*, and *HSP70* determined by qRT-PCR and normalized to the expression of *At4g26410*. Mean ± SEM as well as individual data points from four independent biological repeat experiments are shown. Transcript levels were statistically evaluated for all genotypes within each timepoint by ANOVA followed by Tukey's HSD test ($p < 0.05$). Genotypes are assigned one or more letters based on their statistical group. Genotypes sharing one letter are not significantly different. Source data are available online for this figure.

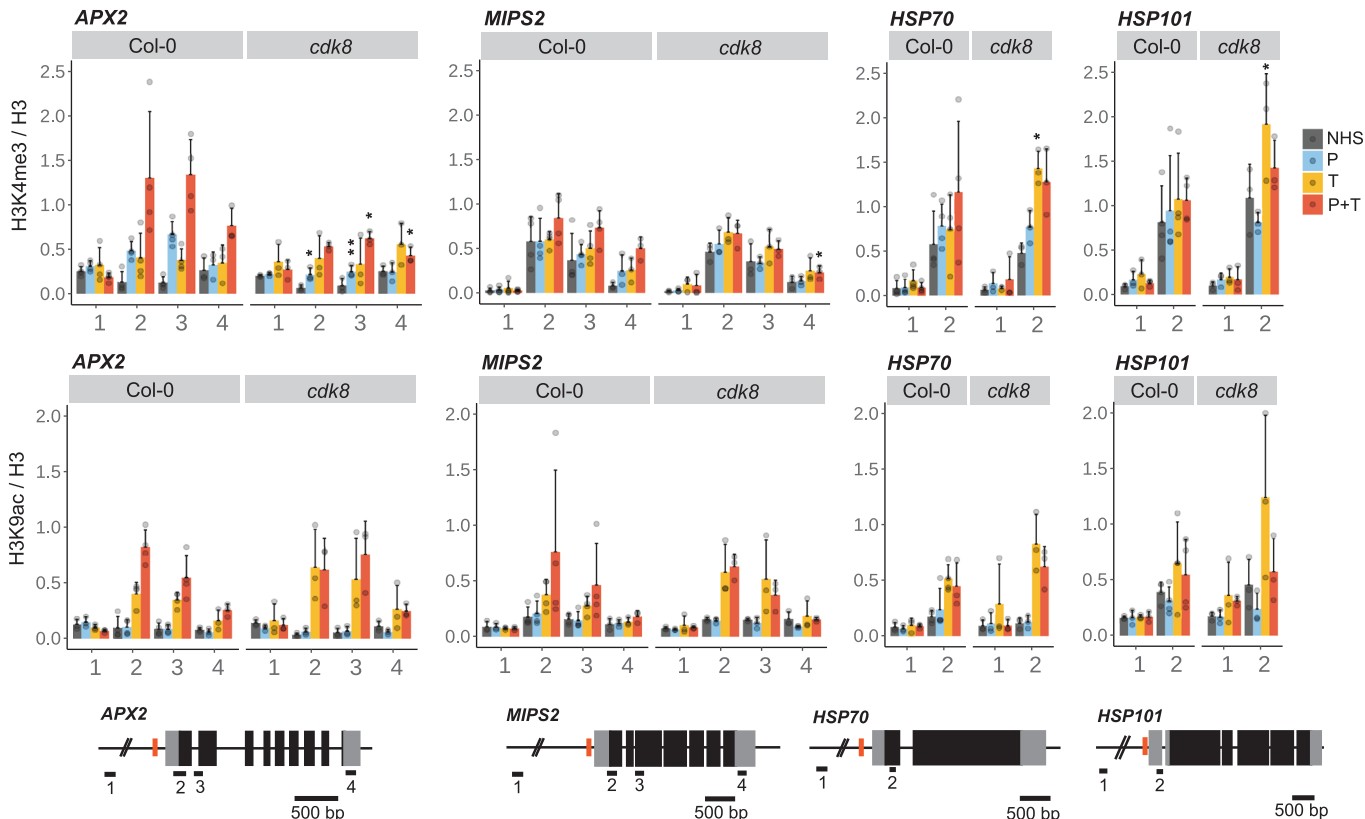

**Figure 7. CDK8 is required for accumulation of H3K4me3 at type II memory gene loci.**

Col-0 wild type and *cdk8* mutant seedlings were subjected to a type II HS regime (cf. Fig. 4A). NHS, no heat stress; P, HS on d 5 only (primed); T, HS on d 7 only (triggered); P + T, HS on both d 5 and d 7 (primed + triggered); each HS consisted of 37 °C for 60 min, and seedlings were sampled on d 7. Enrichment of H3K4me3 (upper) or H3K9ac (lower) relative to histone H3 was determined by ChIP-qPCR. Amplicons targeted the sites indicated on the gene models; red bars indicate binding sites for HSFA2. Data shown are the mean ± SEM ratio of each target relative to H3, as well as individual data points from three or four independent biological repeat experiments. Asterisks denote significant differences in enrichment in *cdk8* relative to Col-0 (*$p < 0.05$; **$p < 0.005$; unpaired, two-tailed *t* test).

no longer induced in the *hsfa1* quadruple mutant (Liu et al, 2011), it is possible that cMed is recruited to these loci via the HSFA2-CDK8 interaction.

Yeast and animal CKM function in part by phosphorylating TFs, thus regulating their activity, localisation and stability (Steinparzer et al, 2019; Alarcon et al, 2009; Fryer et al, 2004). It is an intriguing hypothesis that HSFA2 might be directly phosphorylated by CDK8.

However, our attempts to detect CDK8-dependent phosphorylation on HSFA2 were so far unsuccessful. The interaction between CDK8 and HSFA2 was strongest during and shortly after HS, suggesting that the interaction may occur at the chromatin of common target loci (Fig. 6). This is in line with overlapping binding sites of CDK8 and HSFA2 in the promoters of target genes (Fig. 5, Kappel et al, 2023).

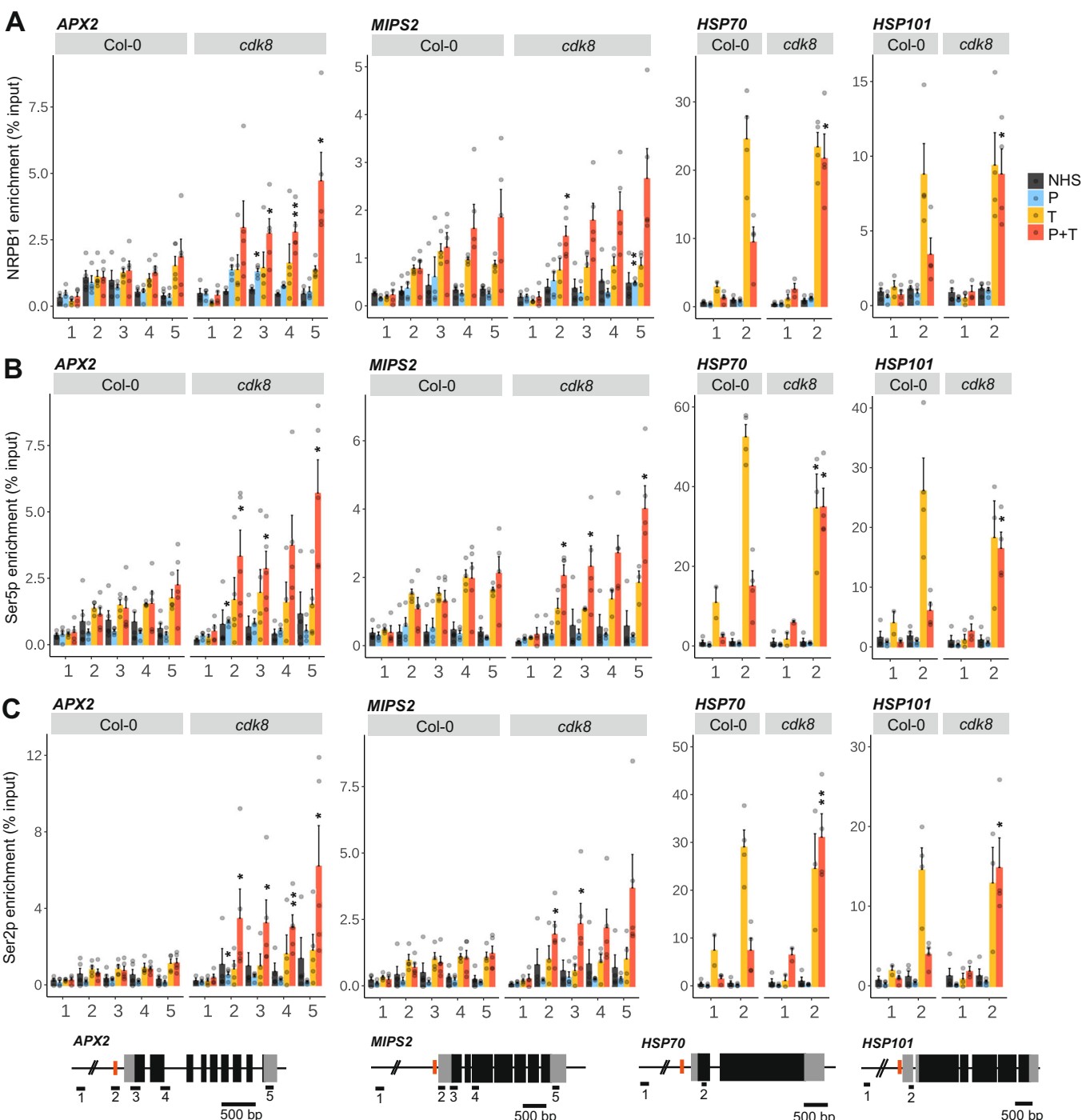

**Figure 8. CDK8 regulates RNA Polymerase II dynamics at type II memory gene loci.**

(A–C) Col-0 wild type and *cdk8* mutant seedlings were subjected to a type II HS regime (cf. Fig. 4A). NHS, no heat stress; P, HS on d 5 only (primed); T, HS on d 7 only (triggered); P + T, HS on both d 5 and d 7 (primed + triggered); each HS consisted of 37 °C for 60 min, and seedlings were sampled on d 7. Occupancy of RNA polymerase II was determined by ChIP-qPCR using antibodies against the major subunit (NRPB1, **A**), the Serine5-phosphorylated (Ser5p, **B**) or the Serine2-phosphorylated forms (Ser2p, **C**) of the C-terminal domain. Amplicons targeted the sites indicated on the gene models; red bars indicate binding sites for HSFA2. Data shown are the mean ± SEM enrichment over input as well as individual data points from four or five independent biological repeat experiments. Asterisks denote significant differences in enrichment in *cdk8* relative to the wild type (*$p < 0.05$; **$p < 0.005$; unpaired, two-tailed *t* test).

CDK8 and cMed were more strongly enriched at memory loci directly after HS in primed (P + T) than unprimed (T) plants (Figs. 5 and EV4). In contrast, neither was enriched in the primed state before recurrent HS (P). Similarly, HSFA2 binds to its target genes in a hit-and-run mode (Lämke et al, 2016). Thus, neither CDK8 nor HSFA2 seem to mark genes for enhanced reactivation during the primed state; rather, modifications to the chromatin, such as hyper-methylation of histone H3K4, appear to prime a locus and make recruitment of CKM-cMed upon recurrent HS more efficient, thus mediating transcriptional memory. During JA-induced priming of dehydration stress genes a partial PIC remained associated with a primed locus (Liu and Avramova, 2016). PIC assembly is a rate-limiting step of transcriptional activation, at least in yeast, and a partial PIC 'scaffold' confers more efficient re-initiation of Pol II (Li et al, 1999; Kuras and Struhl, 1999; Venters and Pugh, 2009; Yudkovsky et al, 2000). Poised or docked Pol II and PIC 'scaffold' components were also implicated in transcriptional memory in yeast and metazoans (Pavri et al, 2005; Light et al, 2013; Maxwell et al, 2014). Whether a partial PIC 'scaffold' is involved in HS-induced transcriptional memory remains to be investigated.

CDK8, MED23 and Pol II binding accumulated towards the 3′-end of genes (Figs. 5 and 8). This was surprising as CDK8 is not assumed to globally travel with Pol II and in fact, CKM and Pol II association with cMed are mutually exclusive in other systems (Knuesel et al, 2009a; Allen and Taatjes, 2015). Pol II binding increased after P + T at 3′-ends, particularly in the absence of CDK8. At the same time, mature transcript levels in *cdk8* were lower, as were Pol II-associated transcripts (Figs. 3, 4, and EV5). Together, this suggests that cMed and CKM travel with Pol II along the gene body. In *cdk8* mutants, unproductive (stalled) Pol II complexes accumulate, similar to a traffic jam. Thus, CDK8 may be required for resolving stalled Pol II complexes, for proper transcription termination and/or for coordinating re-initiation directly after termination. Indeed, in mammalian cells CDK8 is required for pause-release as well as efficient elongation and termination of Pol II (Poss et al, 2016; Galbraith et al, 2013; Donner et al, 2010; Shandilya and Roberts, 2012). Mediator also controls rapid re-initiation events, thus producing 'convoys' of elongating Pol II molecules during transcriptional bursts at highly expressed genes (Tantale et al, 2016; Richter et al, 2022). However, plant genomes lack homologues for a number of protein complexes involved in transcriptional regulation (including NELF and Integrator) so the mechanism of promoter escape, pausing and re-initiation may be quite different (Obermeyer et al, 2023). During transcriptional memory of the *INO1* locus in yeast, CDK8 maintained a partial PIC and hypo-phosphorylated Pol II by preventing re-association of the TFIIK/Kin28 kinase and phosphorylation of Pol II CTD (Light et al, 2013; D'Urso et al, 2016). H3K4me2 was shown to be upstream of PIC formation and required for Pol II recruitment during memory (D'Urso et al, 2016; Sump et al, 2022). During HS memory, H3K4 hyper-methylation was impaired in *cdk8*. Yeast CDK8 modulates the activity of the Set1/COMPASS histone methyltransferase (D'Urso et al, 2016; Law and Finger, 2017); it remains to be investigated whether this interaction is conserved in plants.

The CKM has complex roles in the activation and repression of transcription, particularly in reprogramming gene expression patterns in response to environmental or developmental cues.

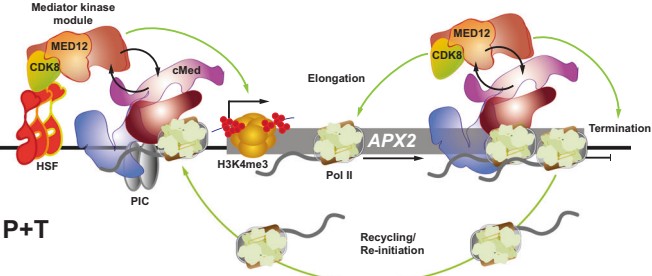

**Figure 9. CDK8 and the CKM regulate HS-induced transcriptional memory at the *APX2* locus promoting efficient transcription by Pol II.**

After a repeated HS (P + T), HSF-containing oligomeric complexes bind to DNA and activate *APX2* transcription. The CKM is recruited by direct interaction with HSFA2/HSFA3. cMed is also recruited along with the CKM to interact with the pre-initiation complex (PIC) and Pol II to initiate transcription. CDK8 promotes hyper-methylation of H3K4 around the TSS. Both cMed and CKM travel with elongating Pol II through the body of the gene, accumulating to the highest levels at its 3′ end. CDK8 may promote aspects of the elongation and termination steps and/or promote efficient recycling/re-initiation of Pol II complexes immediately after termination, thus driving hyper-induced gene expression. Thus, CDK8 inhibits the accumulation of non-productive Pol II at the 3′ end of the gene. Presumably, CKM and Pol II do not directly interact, but rather exist in a rapidly-exchanging equilibrium with cMed.

Transcription repression is mediated independently of kinase activity by blocking the binding of cMed to Pol II in the PIC. In contrast, roles in transcription activation during transcription elongation, termination and re-initiation may require kinase activity, for example through the phosphorylation of gene-specific transcription factors (Donner et al, 2010; Galbraith et al, 2013; Poss et al, 2016). What determines the net effect at individual genes is unknown. Plant CDK8 has been ascribed positive and negative roles in gene expression, however, mechanistic details are lacking. Here, we examined the role of plant CDK8 during stimulus-dependent gene induction and transcriptional memory in unprecedented detail. Besides the expected association of CDK8 with promoter and transcription start site regions, we unexpectedly found CDK8 and cMed enriched at 3′-ends. The temporal and spatial dynamics of HSFA2, CDK8, cMed and Pol II occupancy point to a model where HSFA2 recruits CKM to memory gene loci, where CDK8 promotes enhanced re-induction upon recurrent HS by (1) modifying chromatin environment in the primed state around the TSS, and (2) by promoting highly productive Pol II complexes or Pol II convoys through its roles in elongation, termination, recycling and/or re-initiation (Fig. 9). Our findings enhance our understanding about the function of CDK8 in stress responses and transcriptional stress memory and underline the complex and central role of the Mediator co-regulator complex during transcription regulation.

## Methods

### Plant materials and growth conditions

All *A. thaliana* lines were in the Col-0 background. *pAPX2::LUC* (Liu et al, 2018), *cdk8* (GABI_564F11; Ng et al, 2013; Crawford et al, 2020), *hsp101* (SALK_066374; Wu et al, 2013), *hsfa2-1*

(SALK_008978; Charng et al, 2007), *cct-1*, *cct-2* (SALK_108241) (Gillmor et al, 2010), *med23* (SALK_074015; Dolan et al, 2017), *HSFA2::FLAG-HSFA2* (Friedrich et al, 2021), *NRPB2::NRPB2-FLAG* (Kindgren et al, 2020), *35S::CDK8-MYC* and *35S::CDK8-D176A-MYC* (Zhu et al, 2014) have been described. EMS-induced mutations were backcrossed at least twice before mapping and detailed phenotyping. Seeds were sterilised and grown on 1× MS solid medium with 1% (w/v) glucose. After stratification, seedlings were grown at 23 °C in a 16 h/8 h light/dark cycle in a growth cabinet (white light intensity 120 µE m$^{-2}$ s$^{-1}$). HS treatments were initiated after 5 d of growth.

### Heat stress treatments

To assess basal thermotolerance (bTT), plates with 5 d-old seedlings were incubated at 44 °C for the indicated times. To assess acquired thermotolerance (aTT), 5 d-old seedlings were incubated at 37 °C for 1 h, then 23 °C for 90 min, then 44 °C for 140–240 min. To assess HS memory/maintenance of acquired thermotolerance (maTT), 5 d-old seedlings were first subjected to a acclimatising HS (37 °C for 1 h, 23 °C for 90 min, then 44 °C for 45 min), left to grow for 3 d, and then subjected to 44 °C for 70–120 min. Plates were returned to normal growth conditions for 14 d before analysis (Stief et al, 2014).

### LUC assay

LUC activity was measured using a NightOWL LB983 in vivo imaging system (Berthold Technologies). Seedlings were sprayed with a 1 mM solution of D-luciferin (Promega) in water. Plates were incubated in the dark for 12 min before bioluminescence detection. Using the IndiGO software, images were taken with an exposure time of 200 s. The threshold of LUC was adjusted depending on the experimental setup.

### Construction of transgenic lines

To construct the *pCDK8::CDK8-GSY* line, a 2.7 kb fragment containing the genomic sequences of *CDK8* (no stop codon) and its promoter, and a 783 bp fragment containing the 3′ UTR and flanking sequence, were amplified from Col-0 genomic DNA. The GSY tag (C-terminal orientation) and linker sequence was amplified from the *pEN-R2L3-GSY* plasmid (Besbrugge et al, 2018). The fragments were cloned into an *AscI/NotI*-digested *pGreenII* vector containing the methotrexate resistance gene using InFusion (Takara Bio). Following sequence confirmation, the construct was introduced into *Agrobacterium tumefaciens* GV3101 and transformed into *cdk8* mutant by floral dip (Clough and Bent, 1998). Transgenic individuals were selected on GM media containing 0.1 µg mL$^{-1}$ methotrexate, and checked for single copy insertion. To construct the *pMED23:HA-MED23* line, an InFusion-based approach was utilized. A 1.0 kb fragment containing the *MED23* promoter and an 8.4 kb fragment containing the genomic sequence of *MED23* and the 3′ UTR were amplified from Col-0 genomic DNA. A 3xHA sequence was amplified from pJOG331 plasmid and fused in an N-terminal orientation to the *MED23* gene using InFusion. Fragments were cloned into *pGreenII* as above, and constructs transformed in the *med23* mutant by floral dip; transformants were selected on methotrexate. To construct *35S::FLAG-LUC*, sequences of the *CaMV 35S* promoter, 3×FLAG

tag with linker, firefly LUC and *nos* terminator were amplified and cloned into *pGreenII* with Norflurazone-resistance gene. After floral-dip transformation into the wild-type, transgenic individuals were selected for on GM media containing 1 µM Norflurazone. Oligonucleotide sequences for construct preparation are given in Dataset EV3.

### RNA extraction and qRT-PCR

Seedlings were subjected to either a type I (sustained expression) or type II (altered re-induction) HS treatment as described for each experiment. Samples were snap-frozen in liquid nitrogen and RNA was extracted using the hot phenol method (Friedrich et al, 2021). Genomic DNA was degraded using TURBO™ DNAse (Ambion), and cDNA was reverse-transcribed from 5 µg RNA using SuperScript III (Invitrogen) and oligo-dT primers. 2.5 µL of 1:5 diluted cDNA was used in a 10 µL qPCR reaction using GoTaq qPCR Master Mix (Promega) and a LightCycler 480 (Roche). All data were normalised to the reference transcript *At4g26410* using the comparative Ct method.

### RNA sequencing and analysis

RNA was extracted using the RNeasy Plant Mini kit (Qiagen). Genomic DNA was degraded using TURBO™ DNAse, and RNA quality checked using RNA ScreenTape (TapeStation 4200, Agilent). Library preparation using oligo-dT enrichment of mRNAs, and RNA sequencing were performed by BGI Genomics (http://www.bgi.com) with the DNBseq platform, generating 2 × 150 bp paired-end sequencing reads (~67 million reads per sample). Paired-end reads were mapped against the *A. thaliana* reference genome (TAIR10) using STAR version 2.7.10a (Dobin et al, 2013) with the –quantMode GeneCounts option.

Differential gene expression analysis was performed using the R package DESeq2 (Love et al, 2014). Significantly differentially expressed genes were defined by an adjusted *p*-value of <0.05 and a log$_2$ fold-change of >1 or <−1 for up-regulated or down-regulated genes, respectively, relative to the NHS control for that genotype. In Col-0, +/++ genes were defined as those genes which were significantly up-regulated at T vs. NHS and at P + T vs. P, and which were also up-regulated at P + T vs. T. The 0/+ genes were defined as significantly up-regulated at P + T vs. P, not up-regulated at T vs. NHS, and also up-regulated at P + T vs. T. Venn diagrams were generated using Venny (Oliveros, 2007) and graphs generated using the R packages lattice, latticeExtra and ggplot2.

### Chromatin immunoprecipitation

Chromatin immunoprecipitation was performed essentially as described (Lämke et al, 2016; Kaufmann et al, 2010), with some modifications. Seedlings were subjected to a type II HS regime and harvested immediately after HS. Different cross-linking and ChIP conditions were used depending on the target. For histone modifications, 1–2 g of seedlings were crosslinked in ice-cold MC buffer containing 1% formaldehyde, 2 × 5 min under vacuum (25 mbar). For Pol II and Mediator, 2–3 g seedlings were crosslinked in ice-cold MC buffer containing 1.5 mM ethylene glycol bis(succinimidyl succinate) (EGS; Sigma), 10 min under vacuum. Formaldehyde was added to a

final concentration of 1%, and seedlings were crosslinked for a further 10 min. In both cases, crosslinking was stopped by addition of glycine to a final concentration of 0.125 M and 5 min under vacuum. Seedlings were washed, dried and snap-frozen until analysis. Seedlings were ground under liquid nitrogen, and chromatin was extracted in 25 mL ice-cold, freshly-made M1 buffer supplemented with 1 mM phenyl-methylsulfonylfluoride (PMSF; Roche), 2 mM benzamidine (Sigma), 1× cOmplete Protease Inhibitor Cocktail tablets (Roche), 1/1000 (v/v) Plant Protease Inhibitor Cocktail (Sigma). Extracts were filtered through Miracloth and washed five times by centrifugation in ice-cold, freshly made M2 buffer, then one wash with M3 buffer. Chromatin pellets were resuspended in 1 mL sonication buffer and sonicated for 15, 20 or 25 cycles of 30 s on/30 s off using a Bioruptor (Diagenode) on low-intensity settings. Sonicated chromatin was cleared, mixed with an equal volume of IP buffer, and 100 μL removed as an input control. Chromatin was pre-cleared with 20 μL equilibrated Protein A beads (Invitrogen) for 1 h at 4 °C with 10 rpm rotation. Pre-cleared chromatin was split into equal volumes depending on the targets, and incubated with appropriate primary antibodies overnight at 4 °C with rotation. For histone modifications, 1.5 μg of antibodies against histone H3 (abcam ab1791), H3K4me3 (ab8580), H3K9ac (ab10812), or rabbit IgG (Thermo-Fisher 02-6102) were used. For Pol II, 1.5 μg of antibodies against NRPB1 (Agrisera AS11-1804), CTD repeat YSPTSPS (phospho S5, abcam ab5131), or CTD repeat YSPTSPS (phospho S2, abcam ab5095) were used; rabbit IgG was used as a negative control. For Mediator, 4 μg of antibodies against MYC (abcam ab9106) or HA (Sigma H6908) were used, and rabbit IgG was used as a negative control. Chromatin-antibody complexes were then recovered by incubation with Protein A beads, 3 h at 4 °C with rotation. Following IP, all washing, elution, de-crosslinking, DNA purification and recovery steps were performed as described (Lämke et al, 2016). ChIP DNA was diluted 1/50 (for inputs) or 1/10 (for IPs) and analysed by qPCR using primers listed in Dataset EV3.

### Split-luciferase complementation assay

Proteins of interest were fused to N- and C-terminal fragments of *Renilla* LUC (RLuc) and transiently expressed in tobacco leaves using ß-estradiol-inducible destination vectors pYS40 (RLuc-N) and pYS39 (RLuc-C) (Schatlowski et al, 2010). Vectors were transformed into *A. tumefaciens* GV3101 and transfected into leaves of 4-week-old *N. benthamiana* for transient expression. Expression of the fusion proteins in *N. benthamiana* was induced by brushing infiltrated leaves with 100 μM β-estradiol/0.1% Tween-20 in water. Plants were kept for 24 h in a lid-covered tray. For bioluminescence detection, the leaves were infiltrated with 10 μM ViviRen live cell substrate (Promega). After incubation for 10 min in darkness, bioluminescence was detected with the NightOWL for 30 min.

### Protein co-immunoprecipitation and immunoblotting

In vitro pulldown assay to test the interaction between CDK8 and HSF proteins was performed essentially as described (Friedrich et al, 2021). CDK8 coding sequences were inserted into the pIX-HALO expression vector. pIX-FLAG expression vectors for FLAG-tagged HSFA2 and HSFA3 proteins were previously described (Friedrich et al, 2021). Plasmid combinations were transcribed and co-translated in vitro using TNT wheat germ expression kits (Promega) and immunopurified with MagneHalo™ Beads (Promega). HALO-Tag expressed from empty pIX-HALO vector was used as a negative control. Co-immunoprecipitation of FLAG-tagged proteins was performed essentially as described (Friedrich et al, 2021). Briefly, 1–2 g of seedlings were ground under liquid nitrogen and proteins were extracted in 4 mL non-denaturing buffer containing 50 mM Tris-HCl pH 7.5, 50 mM NaCl, 2% (v/v) Triton X-100 and 5 mM DTT, supplemented with 1 mM phenyl-methylsulfonylfluoride (PMSF; Roche), 2 mM benzamidine (Sigma), 1× cOmplete Protease Inhibitor Cocktail tablets (Roche) and 1/1000 (v/v) Plant Protease Inhibitor Cocktail (Sigma). After 30 min extraction on ice, extracts were cleared by centrifugation. An input sample was removed before extracts were incubated with 50 μL anti-DYKDDDDK paramagnetic beads (Miltenyi Biotec) at 4 °C with rotation for 1.5 h. Extracts were loaded step-wise onto columns and separated at RT using μMACS magnetic isolation kit (Miltenyi Biotec); columns were washed three times with wash buffer (50 mM Tris-HCl pH 7.5, 50 mM NaCl, 0.1% (v/v) Triton X-100, 1× cOmplete Protease Inhibitor) and incubated 10 min on the column with 20 μL native elution buffer (1.5% (v/v) triethylamine, 0.1% (v/v) Triton X-100, 1× protease inhibitors). Native protein complexes were then eluted by pH-shift with native elution buffer, 3×50 μL; eluates were pooled and neutralized by addition of MES pH 3.0 to a final concentration of 180 mM. For analysis by SDS-PAGE, proteins were separated on resolving gels, transferred to nitrocellulose membranes in 15% MeOH/0.03% SDS buffer and immunoblotted using antibodies against FLAG (Sigma F1804; 1:2500), *c*-MYC (Sigma M4439; 1:4000), RNA polymerase II CTD repeat YSPTSPS (phospho S2) (abcam ab5095; 1:1000). Immune-complexed proteins were detected using IRDye secondary antibodies and visualized using an Odyssey Fc imager (LiCOR). Blotted membranes were stained using REVERT total protein stain (LiCOR).

### Analysis of nascent transcription

Plant Native Elongating Transcript-qPCR (plaNET-qPCR) was performed essentially as described, using FLAG-tagged NRPB2 (Kindgren et al, 2020). In all, 2–3 g of 7 d-old seedlings were processed per sample, and *35S::FLAG-LUC* was used as negative control. Native Pol II complexes were immunoprecipitated from total protein extracts using anti-FLAG beads as described for above, except that RNasin® ribonuclease inhibitor (Promega; 1/250 dilution) and 10 mM MgCl$_2$ were added. RNA was extracted from the immunoprecipitated sample using the Hot phenol method, and cDNA synthesized using 6 μM Random Primer Mix (NEB) and SuperScript III. Transcript expression was analysed by qPCR using the primers listed in Dataset EV3; expression was normalized to *ACTIN2*.

## Data availability

RNA-seq data are available at NCBI GEO under accession number GSE232094.

# Peer review information

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

## Acknowledgements

We thank Tesfaye Mengiste (Purdue University, USA), Sebastian Marquardt (University of Copenhagen, DK), Stefan Björklund (Umea University, SWE), Thomas Laux (Freiburg University, DE) and Daniel Schubert (Freie Universität Berlin, DE) for sharing transgenic lines and plasmids. We thank Michael Lenhard for helpful suggestions and Christiane Schmidt and Doreen Mäker for excellent plant care. IB acknowledges funding from the Deutsche Forschungsgemeinschaft (CRC973/project A2, BA3942/5-1) and the European Research Council (ERC CoG 725295 CHROMADAPT).

## Author contributions

**Tim Crawford**: Conceptualization; Data curation; Formal analysis; Investigation; Visualization; Methodology; Writing—original draft; Writing—review and editing. **Lara Siebler**: Conceptualization; Data curation; Formal analysis; Investigation; Visualization; Methodology; Writing—review and editing. **Aleksandra Sulkowska**: Conceptualization; Data curation; Formal analysis; Investigation; Writing—review and editing. **Bryan Nowack**: Conceptualization; Data curation; Formal analysis; Investigation; Visualization; Writing—review and editing. **Li Jiang**: Conceptualization; Data curation; Formal analysis; Investigation; Writing—review and editing. **Yufeng Pan**: Conceptualization; Data curation; Investigation; Writing—review and editing. **Jörn Lämke**: Conceptualization; Resources; Investigation; Methodology; Writing—review and editing. **Christian Kappel**: Data curation; Software; Formal analysis; Investigation; Writing—review and editing.

**Isabel Bäurle**: Conceptualization; Data curation; Formal analysis; Supervision; Funding acquisition; Investigation; Visualization; Writing—original draft; Project administration; Writing—review and editing.

## Funding

## Disclosure and competing interests statement

The authors declare no competing interests.

# Expanded View Figures

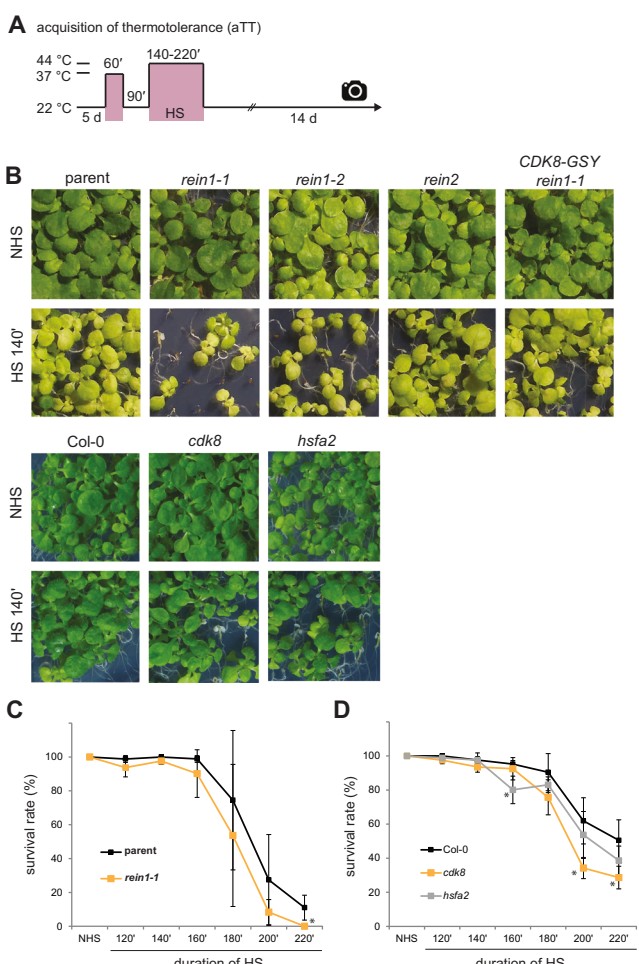

**A** acquisition of thermotolerance (aTT)

**Figure EV1. Acquisition of thermotolerance is slightly impaired in *rein* and *cdk8*.**

(**A**) Treatment scheme for acquired thermotolerance (aTT) assay. 5 d-old seedlings were exposed to 37 °C for 1 h, recovered at 23 °C for 90 min and subsequently exposed to 44 °C for 140–220 min. Images were taken 14 d later. Three or four biological repeat experiments were performed. (**B**) Representative aTT assay with the *rein1-1, rein1-2* and *rein2* mutants, the complementing *CDK8-GSY rein1-1* line and the parental *pAPX2::LUC* line (upper), and the *cdk8* and *hsfa2* mutants and their parental Col-0 wild type (lower). NHS (non-heat stressed) seedlings are shown as controls for normal growth. (**C, D**) Survival rates for *rein1-1* (**C**) and *cdk8, hsfa2* (**D**). Error bars indicate the mean ± SEM of three independent biological replicate experiments. Asterisks denote significant difference of the mutant to the relative control (*$p < 0.05$; unpaired, two-tailed *t* test).

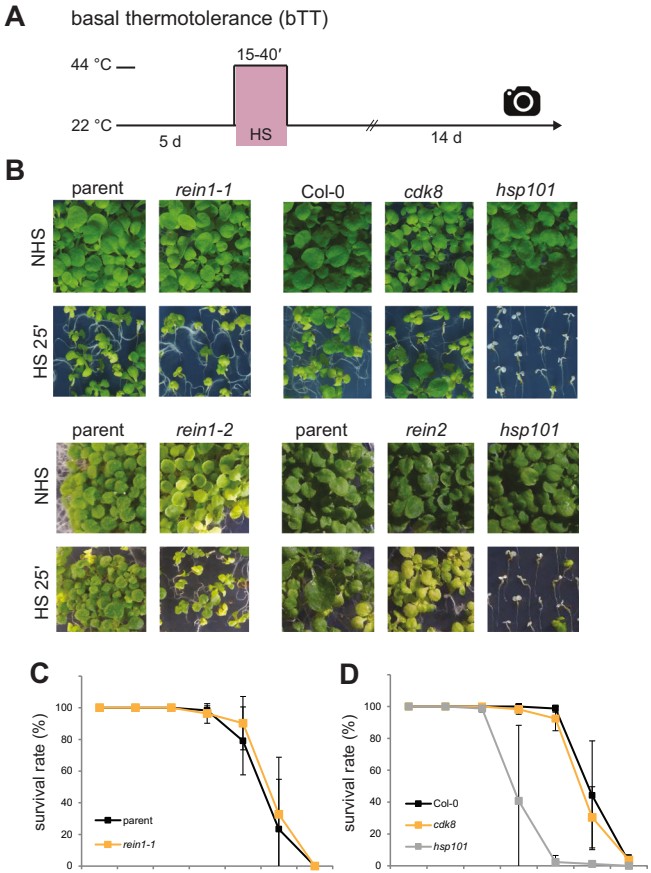

**Figure EV2.   Basal thermotolerance is not affected in *rein* and *cdk8*.**

(A) Treatment scheme for basal thermotolerance (bTT) assays. Five-day-old seedlings were exposed to 44 °C for 15-40 min and images were taken 14 d later. The *hsp101* mutant was included as a control for decreased basal thermotolerance. Three or four biological repeat experiments were performed. (B) Representative bTT assay with *rein1-1* and its parent line (upper left), *cdk8*, *hsp101* and Col-0 wild type (upper right), *rein1-2* and the parent (lower left), and *rein2* (lower right). Each panel shows representative images from plants grown on the same plates. (C, D) Survival rates for *rein1-1* (C) and *cdk8* (D) versus their respective controls Error bars indicate the mean ± SEM of three independent biological replicate experiments.

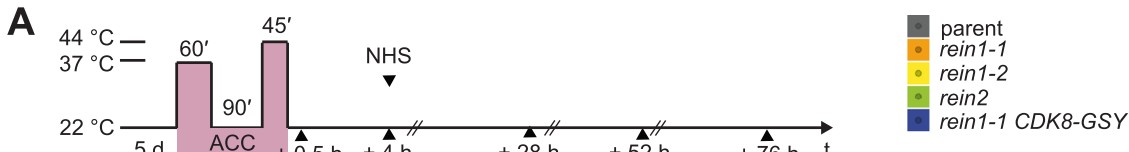

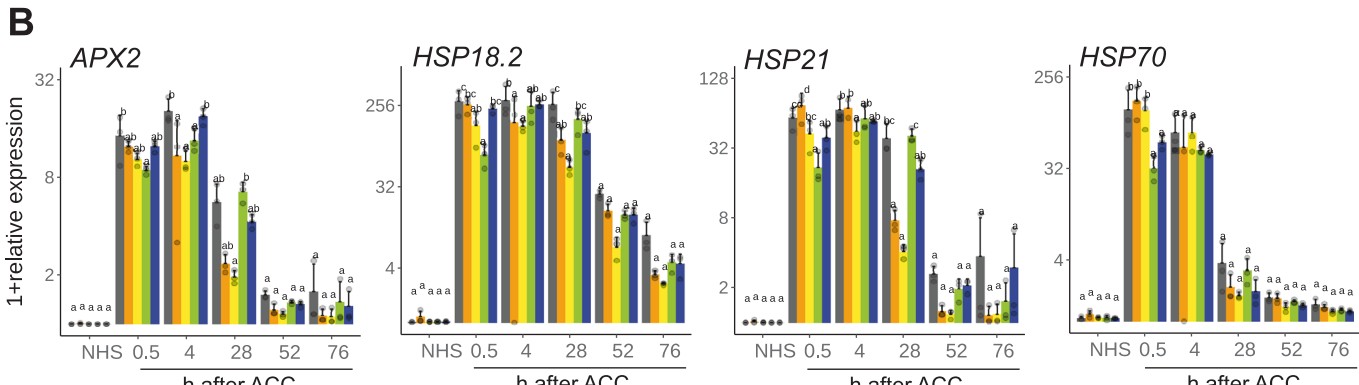

**Figure EV3.  Type I transcriptional HS memory gene expression is impaired in *rein* mutants.**

(A) Treatment schema for the type I HS memory assay (sustained induction). Seedlings were grown for 5 d before being subjected to either a full ACC (37 °C for 60 min, RT for 90 min, 44 °C for 45 min). Seedlings were left to recover and sampled at 0.5, 4, 28, 52 or 76 h after the end of ACC (or 4 h after NHS). (B) Seedlings were treated as indicated and relative transcript levels of three type I memory genes (*APX2*, *HSP18.2*, *HSP21*) and the HS-induced non-memory gene *HSP70* were measured by qRT-PCR and normalized to the expression of *At4g26410*. Data are $\log_2$(1+mean of relative transcript expression) ± SEM of three independent biological replicate experiments, along with individual data points. Transcript levels were statistically evaluated for all genotypes within each timepoint by ANOVA followed by Tukey's HSD test ($p < 0.05$). Genotypes are assigned one or more letters based on their statistical group. Genotypes sharing one letter are not significantly different.

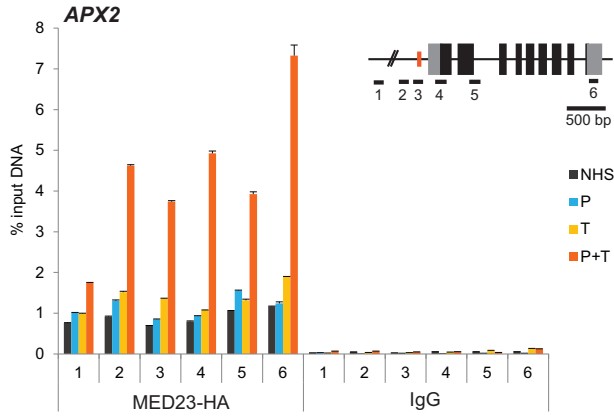

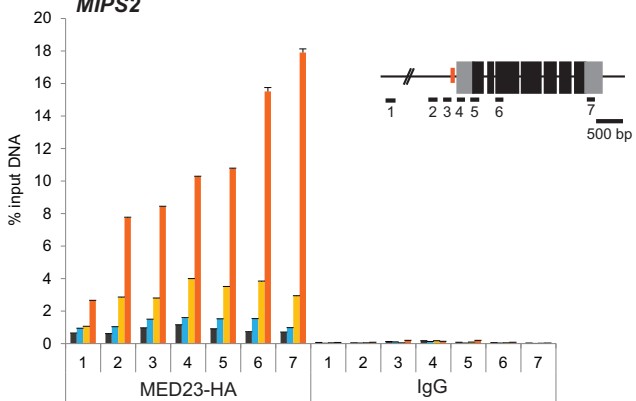

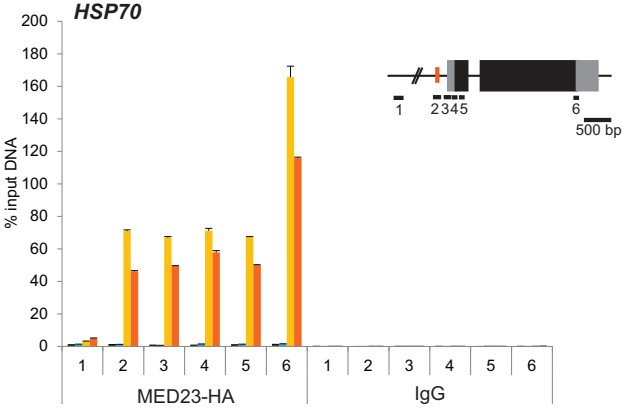

**Figure EV4.  MED23 is recruited to HS memory gene loci during type II memory.**

Seedlings of the MED23-HA line were subjected to a type II HS regime. Occupancy of MED23-HA was determined by ChIP-qPCR using antibodies against the HA tag. Amplicons targeted the sites indicated on the gene models; red bars indicate binding sites for HSFA2. Data from one representative biological replicate are shown ±SEM of three technical replicates; the experiment was repeated with near identical results.

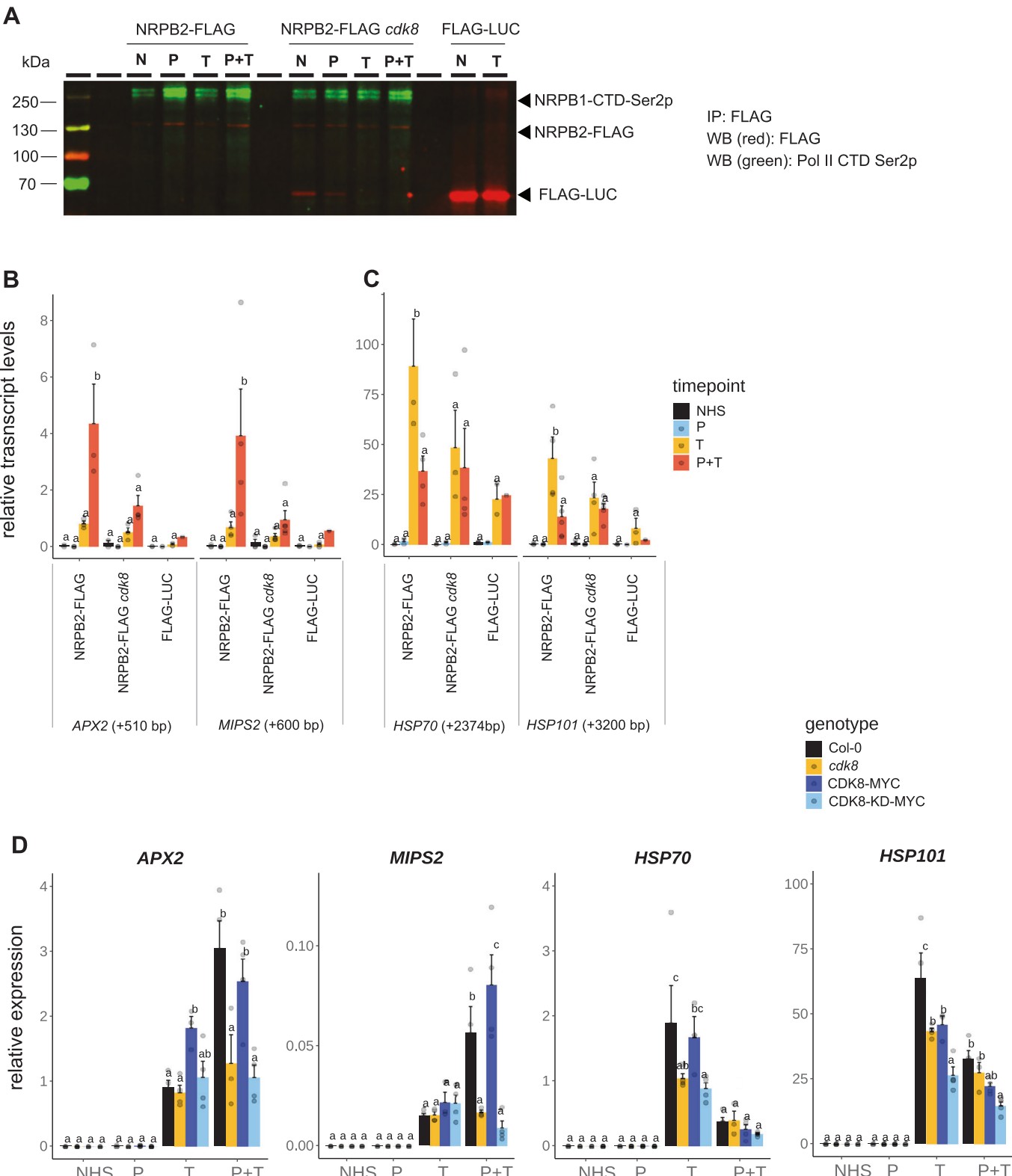

◀  **Figure EV5.  Pol II-associated nascent transcript levels of type II memory gene loci depend on CDK8.**

NRPB2-FLAG seedlings in the wild type or *cdk8* background, and FLAG-LUC control seedlings, were subjected to a type II HS regime (cf. Fig. 4A). NHS, no heat stress; P, HS on d 5 only (primed); T, HS on d 7 only (triggered); P + T, HS on both d 5 and d 7 (primed + triggered); each HS consisted of 37 °C for 60 min, and seedlings were sampled on d 7. (**A**) Native RNA polymerase II complexes were immunoprecipitated from total protein extracts and separated by SDS-PAGE, followed by immunoblotting with antibodies against FLAG (red) or RNA pol II (NRPB1-)CTD-Ser2p (green). (**B**, **C**) Nascent elongating transcript levels in isolated native RNA polymerase II complexes were analysed using plaNET-qPCR. (**D**) Seedlings of Col-0, *cdk8*, CDK8-MYC or CDK8-KD-MYC were subjected to a type II HS regime and relative levels of the unspliced transcripts of *APX2, MIPS2, HSP70* and *HSP101* were measured by qRT-PCR. (**B**, **D**) Data shown are the mean ± SEM of indicated transcript levels, normalized to ACTIN2 (**B**, **C**) or *At4g26410* (**D**), as well as individual data points from at least four independent biological repeat experiments, along with individual data points. Transcript levels were statistically evaluated for all genotypes within each treatment by ANOVA followed by Tukey's HSD test ($p < 0.05$). Genotypes are assigned one or more letters based on their statistical group. Genotypes sharing one letter are not significantly different.

