## [Peer Review File · The EMBO Journal]

The mediator kinase module enhances polymerase activity to regulate transcriptional memory after heat stress in Arabidopsis

Tim Crawford, Lara Siebler, Aleksandra Sulkowska, Bryan Nowack, Li Jiang, Yufeng Pan, Jörn Lämke, Christian Kappel, and Isabel Bäurle

DOI: [10.15252/embj.2023114621](https://doi.org/10.15252/embj.2023114621)

Corresponding author(s): Isabel Bäurle (isabel.baeurle@uni-potsdam.de)

Review Timeline:

Submission Date:	26th May 23
Editorial Decision:	9th Jul 23
Revision Received:	28th Nov 23
Editorial Decision:	7th Dec 23
Revision Received:	12th Dec 23
Accepted:	14th Dec 23

Editor: William Teale

Transaction Report:

Dear Isabel,

Thank you again for the submission of your manuscript entitled "Mediator kinase module enhances polymerase activity to regulate transcriptional memory after heat stress" and for your patience during the review process. We have now received the reports from the referees, which I copy below.

As you can see from their comments, all referees are broadly supportive of your manuscript being published in The EMBO Journal. Referees 2 and 3 raise technical questions about the quality of individual datasets and/or their analysis. Please consider these points carefully and decide whether some extra experimentation would be necessary fully to support your conclusions. Also, feel free to contact me if you would like to go over these points via Zoom, or have any questions, need further input on the referee comments, or if you anticipate any problems in addressing them.

Based on the overall interest expressed in the reports, therefore, I would like to invite you to address the comments of all referees in a revised version of the manuscript. Please accompany this revised manuscript with a point-by-point response to the reviewers. Should you decide that the suggested extra experiments are not necessary, final acceptance will depend on the referees accepting your argumentation. I should add that it is The EMBO Journal policy to allow only a single major round of revision and that it is therefore important to resolve these concerns at this stage. Please follow the instructions below when preparing your manuscript for resubmission.

I would also like to point out that as a matter of policy, competing manuscripts published during this period will not be taken into consideration in our assessment of the novelty presented by your study ("scooping" protection). We have extended this 'scooping protection policy' beyond the usual 3 month revision timeline to cover the period required for a full revision to address the essential experimental issues. Please contact me if you see a paper with related content published elsewhere to discuss the appropriate course of action.

Again, please contact me at any time during revision if you need any help or have further questions.

Thank you very much again for the opportunity to consider your work for publication. I look forward to your revision.

Best regards,

William

William Teale, Ph.D.
Editor
The EMBO Journal

When submitting your revised manuscript, please carefully review the instructions below and include the following items:

- 1) a .docx formatted version of the manuscript text (including legends for main figures, EV figures and tables). Please make sure that the changes are highlighted to be clearly visible.
- 2) individual production quality figure files as .eps, .tif, .jpg (one file per figure).
- 3) a .docx formatted letter INCLUDING the reviewers' reports and your detailed point-by-point response to their comments. As part of the EMBO Press transparent editorial process, the point-by-point response is part of the Review Process File (RPF), which will be published alongside your paper.
- 4) a complete author checklist, which you can download from our author guidelines ([https://wol-prod-cdn.literatumonline.com/pb-assets/embo-site/Author Checklist%20-%20EMBO%20J-1561436015657.xlsx](https://wol-prod-cdn.literatumonline.com/pb-assets/embo-site/Author%20Checklist%20-%20EMBO%20J-1561436015657.xlsx)). Please insert information in the checklist that is also reflected in the manuscript. The completed author checklist will also be part of the RPF.
- 5) Please note that all corresponding authors are required to supply an ORCID ID for their name upon submission of a revised manuscript.
- 6) We require a 'Data Availability' section after the Materials and Methods. Before submitting your revision, primary datasets

produced in this study need to be deposited in an appropriate public database, and the accession numbers and database listed under 'Data Availability'. Please remember to provide a reviewer password if the datasets are not yet public (see <https://www.embopress.org/page/journal/14602075/authorguide#datadeposition>). If no data deposition in external databases is needed for this paper, please then state in this section: This study includes no data deposited in external repositories. Note that the Data Availability Section is restricted to new primary data that are part of this study.

Note - All links should resolve to a page where the data can be accessed.

8) For data quantification: please specify the name of the statistical test used to generate error bars and P values, the number (n) of independent experiments (specify technical or biological replicates) underlying each data point and the test used to calculate p-values in each figure legend. The figure legends should contain a basic description of n, P and the test applied. Graphs must include a description of the bars and the error bars (s.d., s.e.m.).

9) We would also encourage you to include the source data for figure panels that show essential data. Numerical data can be provided as individual .xls or .csv files (including a tab describing the data). For 'blots' or microscopy, uncropped images should be submitted (using a zip archive or a single pdf per main figure if multiple images need to be supplied for one panel). Additional information on source data and instruction on how to label the files are available at .

10) We replaced Supplementary Information with Expanded View (EV) Figures and Tables that are collapsible/expandable online (see examples in <https://www.embopress.org/doi/10.15252/embj.201695874>). A maximum of 5 EV Figures can be typeset. EV Figures should be cited as 'Figure EV1, Figure EV2" etc. in the text and their respective legends should be included in the main text after the legends of regular figures.

12) Our journal encourages inclusion of *data citations in the reference list* to directly cite datasets that were re-used and obtained from public databases. Data citations in the article text are distinct from normal bibliographical citations and should directly link to the database records from which the data can be accessed. In the main text, data citations are formatted as follows: "Data ref: Smith et al, 2001" or "Data ref: NCBI Sequence Read Archive PRJNA342805, 2017". In the Reference list, data citations must be labeled with "[DATASET]". A data reference must provide the database name, accession number/identifiers and a resolvable link to the landing page from which the data can be accessed at the end of the reference. Further instructions are available at .

Additional instructions for preparing your revised manuscript:

At EMBO Press we ask authors to provide source data for the main manuscript figures. Our source data coordinator will contact

you to discuss which figure panels we would need source data for and will also provide you with helpful tips on how to upload and organize the files.

We realize that it is difficult to revise to a specific deadline. In the interest of protecting the conceptual advance provided by the work, we recommend a revision within 3 months (7th Oct 2023). Please discuss the revision progress ahead of this time with the editor if you require more time to complete the revisions. Use the link below to submit your revision:

Referee #1:

In this manuscript Crawford et al carry out a forward genetic mutant screen aimed at identifying factors involved in heat-stress memory in Arabidopsis. The authors describe three mutants that were identified in this screen, rein1 to rein3. Rein1 and rein2 were both mutated in the same CDK8 gene. Rein3 was found to be defective in MED12. Genetic analyses and mutant rescuing with a CDK8 transgene confirm the identity of rein1 and rein2 as mutant alleles of CDK8. The authors then find that the identified mutants are affected in physiological HS memory. To study the effects at the molecular level, the authors then perform qPCR and RNA-seq showing that CDK8 defective plants are affected in transcriptional HS memory, very much like hsa2. Moving to a set of elegant biochemical analyses, the authors then demonstrate that CDK8 binds to HS memory genes and requires the direct interaction with HSFA2 to perform this activity. Furthermore, the authors show that this activity is dependent on the kinase domain of CDK8. Analyses at the chromatin level suggest that CDK8 may be involved in H3K4me3 accumulation at tested HS memory genes. Finally, the authors show that CDK8 is required for efficient transcription by Pol II.

This manuscript provides key novel insights into the molecular mechanisms of HS memory that can be of interest to a broad audience. The manuscript is well written and overall, most of the conclusions are supported by strong experimental data. My only criticism concerns the data on MED12 that is supported by less strong evidence than CDK8 (e.g. missing complementation by transgene). Maybe the authors could tone down the role of MED12 in the discussion and the proposed model? The MED12 mutant, even though it may only be a weak allele, does not show the same effects as the CDK8 mutant.

Here are my minor comments:

To use proper nomenclature, the mutants should be called rein1-1 rein1-2 and rein2 (instead of rein1, rein2 and rein3). Also, the authors should not use rein1-3 to designate the three mutants, it is misleading for geneticists as this designation could be understood as the third mutant allele of rein1.

Line 132 and following: Was a complementation cross done between mutants? What was the result of that? Have there been any backcrosses and were these showing that the mutations are recessive?

Line 134: Is the pAPX2::LUC transgene also present in the cdk8 mutant that was used for the crosses? Please specify that the authors analyzed the F1 generation of the crosses.

Line 145: How many times were the rein1-rein3 mutants backcrossed to the reporter line before carrying out the detailed studies. Backcrosses are important in order to remove unrelated EMS mutations.

Line 163: Provide a reference for the source of the type II memory genes identification.

Line 165: rein3 seems to affect a different set of genes with one gene overlap, a point to discuss? Would one not expect LUC transcript levels to be affected in rein3? This should be discussed.

Line 169: Refer to the figures that show the APX2, HSP18.2, and HSP21 expression data.

Why was rein3 not also rescued by transgene approach like it was done for rein1 and rein2?

Line 210: CDK8 levels do not seem to be constant, but rather decrease over time (WB:MYC on the blot).

Figure 6: In 6B there is FLAG-HSFA2 signal in the in the IP of the HALO tag only (should be a negative control), could this be non-specific binding? How often was this experiment reproduced?

Line 243: The complete paragraph seems to be unrelated to the presented study. In introductory sentence as to why the authors move to these factors and why it is important will help the reader. Is there a relation between MED23 and MED12, why not focus on and test MED12?

Overall, in the discussion, please refer back to your figures when mentioning specific results, this will further strengthen the argumentations. I would suggest to the authors to be a bit more specific in the discussion, as it is now it seems to be at a very high altitude not linked enough to the presented data.

A general question that could be discussed: As CDK8 is enriched in the 3' end and Pol II is enriched in the 3' end of genes in cdk8, could it be that CDK8 prevents Pol II stalling (removing/preventing the traffic jam) at the 3' end rendering transcription more productive?

Referee #2:

This study identifies a critical role for the Mediator complex in priming plants to respond to recurring heat stress through a mechanism known as transcriptional memory. The authors used a clever genetic screening approach based on a bioluminescent reporter to identify mutations in two Mediator subunits, Med12 and Cdk8, that impair induction of a reporter gene after recurrent heat shock. They convincingly show using qPCR and RNA-seq that Cdk8 and Med12, which are both part of the Mediator kinase module, are necessary for induction of a set of genes that are induced (or hyper-induced) after recurrent heat shock. They also show that the kinase activity of Cdk8 is required for this induction, and that Cdk8 recruitment to these genes requires the heat shock transcription factor HSFA2, which it binds in vivo. HSFA2 has been previously shown to be required for this heat shock transcriptional memory, and is itself induced by heat shock, so this recruitment mechanism seems reasonable. The author's identification of the Mediator kinase module as the key co-regulator in HSFA2-directed transcriptional memory is an important finding, and I think their efforts to understand how the Mediator kinase module regulates transcription of these genes are promising- although I do have a couple of concerns about interpretation of some of their ChIP-qPCR experiments that I will outline below. Despite these and a few other minor concerns, I still think that this is one of the more in-depth mechanistic studies on how Mediator and its kinase module functions in plants and the authors' findings would be of broad interest. This system for heat shock transcriptional memory in plants also a powerful way of looking at a specific function for the Mediator kinase module, and I think this has a lot of promise as a system to study the mechanisms involved in Mediator's role in transcriptional memory.

Moderate points:

1. The enrichment of Cdk8 over the gene body and towards the 3' end of the gene is a very interesting observation, and this together with the increased levels of Pol II in the cdk8 mutant do provide some important clues about the transcriptional mechanism involved. However, I'm moderately concerned about these particular results given that it's unclear if the authors had sufficient resolution in their ChIP-qPCR experiments to distinguish these regions of the gene. This is a concern because H3K4m3 and to a lesser extent H3K9ac are showing up on the 3' ends of a couple of the genes shown in Fig7. I also wouldn't expect both Ser5 and Ser2 forms of Pol II to be equally distributed across the gene body (Fig8). What were the average fragment sizes after sonication? Given the importance of these results, it could be important to revisit a couple of these key experiments (particularly the Cdk8 binding and Pol II) using a technique that has better resolution - CUT&RUN or CUT&Tag could be good options here. I also think that some of the conclusions regarding the differences observed by ChIP-seq should be re-evaluated given my comments below (see minor point #3). For example, some of the conclusions in the text eg line 275-276 "MIPS, this enrichment was further increasedbut less so in cdk8" are not supported by the data shown even with the current statistical analysis.

2. Are the differences in PolII-associated nascent transcripts between WT and cdk8 mutant significant (Fig. EV6)? These data look quite variable and I didn't see any statistical analysis in that figure. I think this is reasonably important in terms of mechanism.

Minor points:

1. Are the binding sites/motifs for HSFA2 known, and if so - can these be shown on the ChIP-qPCR schematics relative to the primers used for the analysis. Based on the pAPX2::LUC reporter used - I presumed the HSFA2 binding sites are in the

promoter fragment, but this was not clear from the text. This is important given the observations about Mediator/Cdk8 binding at the 3' end of the gene targets analyzed.

2. The Abcam H3M4me3 antibody used for ChIP-qPCR is unfortunately not particularly specific for me3 vs me2 (see <https://doi.org/10.1016/j.molcel.2018.08.015>). I don't think this is problematic for the data shown here, but the authors should probably interpret their findings as reflecting some combination of H3K4me2 and me3 enrichment.

3. Most of the qPCR data (ChIP and RT) are shown with error bars as SEM, but I think SDEV would be more appropriate given the sample sizes. In addition, the authors should consider whether t-tests are appropriate for every comparison given that multiple genotypes are being compared. This is a relatively minor point, but some of these data is quite variable (as is typical for some of these experiments) and a more stringent criteria for identifying significant differences would help to focus on those that are most biologically relevant. A very minor point is that it's difficult to see the data points and error bars when bars are colored black, so selecting a different color for the wild-type controls would be helpful.

Optional point/question for authors:

4. Is there a super-elongation complex that functions in plants much like it does in mammalian cells? Although you discuss pausing/release as a potential mechanism in the discussion, I was unsure if Pol II pausing/release work in the same way in plants as it does in animals. A brief outline of what's known in plants about these processes could be helpful to readers who are working in other systems.

Referee #3:

Crawford et al performed a genetic screen to identify regulators of heat-stress induced transcriptional memory (i.e. priming) in the model plant *Arabidopsis thaliana*. They identified the genes MED12 and CDK8, which are components of the dynamic module (CKM) of mediator transcriptional co-activator complex. Based in these findings, they carry out a comprehensive molecular and biochemical characterization of the role of CDK8 and MED12 in HS transcriptional memory. They demonstrate that HS induced the recruitment of CDK8 to target genes by the transcription factor HSFA2, and confirmed direct interaction between CDK8 and HSFA2 in vivo. They further show that CDK8 interacts with cMed, and is required for HS-induced H3K9me3 at the HS memory gene APX2, but surprisingly not at MIPS2 gene (another HS-memory gene). Last, they found that repeated HS leads to the binding of CDK8 along gene body of memory genes, potentially modulating the PolIII transcriptional activity. Intriguingly, in the absence of CDK8, non-productive PolIII accumulate at HS memory genes.

This is certainly an interesting work reporting novel finding that shed light on the role of CKM in the establishment of active chromatin states required for short-term transcriptional memory in response to HS. The manuscript is well written and results are presented with clarity and sufficient details.

I have nonetheless two concerns:

Lines 263-2. It is surprising that despite CDK8 binds to APX2 and MIPS2 gene locus, and is required for their transcriptional memory, *cdk8* mutant only affects H3K9me3 at APX2 (Figure 7). Indeed, MIPS2 has a very slightly decrease at only one probe, which given the large number of non-independent statistical tests performed can be by pure chance. The seemingly gene-specific function of CDK8 in H3K4me3 deposition at HS-memory genes weaken the statement "CDK8 is required for H3K9me3 accumulation at HS memory gene loci". The authors might want to consider to perform ChIP-seq experiment to evaluate the set of genes with varying H3K9me3 levels in wt and *cdk8* mutants in response to P+T. A

Abstract: "CDK8 also binds to the 3' region of target genes, where it promotes elongation, termination or rapid initialisation of Pol II complex" It is unclear from the presented results whether specific binding to 3' region has any role on any of the activities mentioned here. Indeed, 3' binding is accompanied with gene body and promoter binding, making it impossible to assign a role to any specific genic region. Also, Figure 8 clearly shows that the lack of CDK8 increases the occupancy of total as well as elongating PolIII, contradicting the conclusion "promotes elongation, termination or rapid initialisation of Pol II complex" These statements in the abstract should be qualified, or else supported by direct evidences showing a role for CDK8 on promoting elongation, termination or rapid initialisation, such as provided in

Referee #1:

In this manuscript Crawford et al carry out a forward genetic mutant screen aimed at identifying factors involved in heat-stress memory in Arabidopsis. The authors describe three mutants that were identified in this screen, rein1 to rein3. Rein1 and rein2 were both mutated in the same CDK8 gene. Rein3 was found to be defective in MED12. Genetic analyses and mutant rescuing with a CDK8 transgene confirm the identity of rein1 and rein2 as mutant alleles of CDK8. The authors then find that the identified mutants are affected in physiological HS memory. To study the effects at the molecular level, the authors then perform qPCR and RNA-seq showing that CDK8 defective plants are affected in transcriptional HS memory, very much like hsf2. Moving to a set of elegant biochemical analyses, the authors then demonstrate that CDK8 binds to HS memory genes and requires the direct interaction with HSFA2 to perform this activity. Furthermore, the authors show that this activity is dependent on the kinase domain of CDK8. Analyses at the chromatin level suggest that CDK8 may be involved in H3K4me3 accumulation at tested HS memory genes. Finally, the authors show that CDK8 is required for efficient transcription by Pol II.

This manuscript provides key novel insights into the molecular mechanisms of HS memory that can be of interest to a broad audience. The manuscript is well written and overall, most of the conclusions are supported by strong experimental data.

My only criticism concerns the data on MED12 that is supported by less strong evidence than CDK8 (e.g. missing complementation by transgene). Maybe the authors could tone down the role of MED12 in the discussion and the proposed model? The MED12 mutant, even though it may only be a weak allele, does not show the same effects as the CDK8 mutant.

>> We thank the reviewer for his comments. We show that rein2 (med12) is defective in physiological HS memory, and that a second med12 allele (cct-2) shows the same phenotype. Unfortunately, we were not successful in cloning a construct for transgenic complementation, possibly due to the large size of the MED12 gene (9 kb). When revising the manuscript, we noted a mistake in the data for rein2 in previous Figure 3. We have repeated the experiment in 3 biological replicates and find an expression phenotype similar to rein1, albeit somewhat weaker. We confirmed these findings in a separate time course experiment for hyperinduction of gene expression (Appendix Fig S2). This finding is consistent with the physiological phenotype and both proteins acting together in a complex. Notably, rein2 is likely a hypomorphic mutant, as it contains an amino acid substitution in MED12, while rein1-1 and rein1-2 are likely full loss of function mutants caused by premature stop codons, and rein2 may therefore show a weaker phenotype.

Here are my minor comments:

To use proper nomenclature, the mutants should be called rein1-1 rein1-2 and rein2 (instead of rein1, rein2 and rein3). Also, the authors should not use rein1-3 to designate the three mutants, it is misleading for geneticists as this designation could be understood as the third mutant allele of rein1.

>> We have changed the names of the mutants to rein1-1, rein1-2 and rein2 throughout.

Line 132 and following: Was a complementation cross done between mutants? What was the result of that? Have there been any backcrosses and were these showing that the mutations are recessive?

>> We have added the data on a cross between rein1-1 and rein1-2, which does not complement the mutant phenotype, supporting that the two mutations are allelic (Appendix Fig S1C). At least two backcrosses were performed before detailed phenotyping. The mutations behave recessively, as can be seen for example from the dominance of the wild-type allele in all crossings with wild type/parent

(Fig 1D, Appendix Fig S1, l. 135-139). For *rein1-1*, the transgenic complementation also confirms that *rein1-1* behaves recessive.

Line 134: Is the pAPX2::LUC transgene also present in the *cdk8* mutant that was used for the crosses? Please specify that the authors analyzed the F1 generation of the crosses.

>> *We analyzed the F1 generation. We have specified this information in the text and legend. The pAPX2::LUC transgene was not present in the cdk8 T-DNA mutant.*

Line 145: How many times were the *rein1-rein3* mutants backcrossed to the reporter line before carrying out the detailed studies. Backcrosses are important in order to remove unrelated EMS mutations.

>> *All mutants were backcrossed at least twice before mapping and detailed phenotypic analysis (cf. l. 443-444). Mutant identification was in the F2 of the second backcross, based on LUC signal after P+T. For all three mutants, a clear mapping position was identified. Whenever possible, mutant phenotypes were confirmed in at least two independent alleles.*

Line 163: Provide a reference for the source of the type II memory genes identification.

>> *We have added the reference Liu et al 2018 in l. 166.*

Line 165: *rein3* seems to affect a different set of genes with one gene overlap, a point to discuss? Would one not expect LUC transcript levels to be affected in *rein3*? This should be discussed.

>> *See above (first comment); we have repeated the expression analysis (new Fig. 3, Appendix Fig S2); the *rein2/med12* phenotype is in fact similar to that of *rein1-1* and *rein1-2*, e. g. LUC transcript levels are affected, l. 168-173.*

Line 169: Refer to the figures that show the APX2, HSP18.2, and HSP21 expression data.

>> *We have added the reference to the respective Figure (Fig. EV3A-B, l.175).*

Why was *rein3* not also rescued by transgene approach like it was done for *rein1* and *rein2*?

>> *MED12 is a very large gene with coding sequence of 6 kb and genomic coding sequence of 9 kb. So far, we have unfortunately not succeeded in cloning a complementation construct.*

Line 210: CDK8 levels do not seem to be constant, but rather decrease over time (WB:MYC on the blot).

>> *The input (lower panel) in Fig. 6c shows roughly constant CDK8-MYC levels, as would be expected from a construct that is driven by the 35S promoter. Thus, changes in binding to target loci are unlikely to be caused by changed protein levels. We have modified this sentence for clarity, l. 215-217.*

Figure 6: In 6B there is FLAG-HSFA2 signal in the in the IP of the HALO tag only (should be a negative control), could this be non-specific binding? How often was this experiment reproduced?

>> *We have performed this pull-down experiment twice independently with very similar results. Both experiments show faint background, as is very often found for this type of experiment. Importantly, we go on to confirm the interaction in stably transformed Arabidopsis lines (Fig. 6C).*

Line 243: The complete paragraph seems to be unrelated to the presented study. In introductory

sentence as to why the authors move to these factors and why it is important will help the reader. Is there a relation between MED23 and MED12, why not focus on and test MED12?

>> *CKM and core Mediator do not necessarily occupy the same genomic sites, as they can act independently. We have clarified the motivation of investigating a subunit of the core Mediator in l. 252-253.*

Overall, in the discussion, please refer back to your figures when mentioning specific results, this will further strengthen the argumentations. I would suggest to the authors to be a bit more specific in the discussion, as it is now it seems to be at a very high altitude not linked enough to the presented data.

>> *We have introduced referrals to the Figures and clarified the discussion.*

A general question that could be discussed: As CDK8 is enriched in the 3' end and Pol II is enriched in the 3' end of genes in *cdk8*, could it be that CDK8 prevents Pol II stalling (removing/preventing the traffic jam) at the 3' end rendering transcription more productive?

>> *We thank the reviewer for their suggestion; this hypothesis forms part of our model (cf. l. 430-431).*

Referee #2:

This study identifies a critical role for the Mediator complex in priming plants to respond to recurring heat stress through a mechanism known as transcriptional memory. The authors used a clever genetic screening approach based on a bioluminescent reporter to identify mutations in two Mediator subunits, Med12 and Cdk8, that impair induction of a reporter gene after recurrent heat shock. They convincingly show using qPCR and RNA-seq that Cdk8 and Med12, which are both part of the Mediator kinase module, are necessary for induction of a set of genes that are induced (or hyper-induced) after recurrent heat shock. They also show that the kinase activity of Cdk8 is required for this induction, and that Cdk8 recruitment to these genes requires the heat shock transcription factor HSFA2, which it binds *in vivo*. HSFA2 has been previously shown to be required for this heat shock transcriptional memory, and is itself induced by heat shock, so this recruitment mechanism seems reasonable. The author's identification of the Mediator kinase module as the key co-regulator in HSFA2-directed transcriptional memory is an important finding, and I think their efforts to understand how the Mediator kinase module regulates transcription of these genes are promising- although I do have a couple of concerns about interpretation of some of their CHIP-qPCR experiments that I will outline below. Despite these and a few other minor concerns, I still think that this is one of the more in-depth mechanistic studies on how Mediator and its kinase module functions in plants and the authors' findings would be of broad interest. This system for heat shock transcriptional memory in plants also a powerful way of looking at a specific function for the Mediator kinase module, and I think this has a lot of promise as a system to study the mechanisms involved in Mediator's role in transcriptional memory.

Moderate points:

1. The enrichment of Cdk8 over the gene body and towards the 3' end of the gene is a very interesting observation, and this together with the increased levels of Pol II in the *cdk8* mutant do provide some important clues about the transcriptional mechanism involved. However, I'm moderately concerned about these particular results given that it's unclear if the authors had sufficient resolution in their CHIP-qPCR experiments to distinguish these regions of the gene. This is a concern because H3K4m3 and to a lesser extent H3K9ac are showing up on the 3' ends of a couple of the genes shown in Fig7. I also wouldn't expect both Ser5 and Ser2 forms of Pol II to be equally

distributed across the gene body (Fig8). What were the average fragment sizes after sonication? Given the importance of these results, it could be important to revisit a couple of these key experiments (particularly the Cdk8 binding and Pol II) using a technique that has better resolution - CUT&RUN or CUT&Tag could be good options here. I also think that some of the conclusions regarding the differences observed by ChIP-seq should be re-evaluated given my comments below (see minor point #3). For example, some of the conclusions in the text eg line 275-276 "MIPS, this enrichment was further increasedbut less so in cdk8" are not supported by the data shown even with the current statistical analysis.

>>We thank the reviewer for their helpful comments. The chromatin was sheared to 300-500 bp (example gel attached below). For CDK8 and Pol II at APX2 and MIPS2, we see clear enrichment in the 3'-most amplicon, relative to the closest amplicons, which are 1.7 and 1.0 kb away, respectively. Similarly, the negative control amplicon for each gene is in the far 5'-region, at least 3 kb away and they lack enrichment. In contrast, for the histone modifications, the enrichment at the 3'-most amplicons in APX2 and MIPS2 is lower than that for the amplicon closest to the TSS. We have tried to establish CUT&RUN and CUT&Tag under our conditions, but unfortunately failed to establish a working protocol. Also, to our knowledge, CUT&RUN in Arabidopsis has so far been reported for proteins that bind very tightly to DNA, such as histone modifications, but not for proteins that are only loosely bound to the chromatin such as CDK8 and Pol II. The distribution of Ser2p and Ser5p-modified Pol II that we observed in our ChIP-qPCR experiments is consistent with the genome-wide average distribution that has been reported in plants in a number of publications (see e. g. Obermeyer et al. (2023) Nucleic Acids Res. 10.1093/nar/gkad825]; Zhu et al. (2018) Nature Plants 10.1038/s41477-018-0280-0; Kindgren et al. (2020) Nucleic Acids Res. 10.1093/nar/gkz1189). The evidence suggests that the differences between Ser2p and Ser5p Pol II distributions in plants are much smaller than those observed in animals, and the overall profile is quite different, with an even distribution across most of the gene body and a 3' peak of Ser2p (more pronounced) and Ser5p (less pronounced). The typical accumulation of Ser5p near the TSS is seen in yeast and humans, but appears absent in Arabidopsis (reviewed in Obermeyer et al. (2023) Plant Journal 10.1111/tpj.16115). Plant genomes lack homologues for a number of protein complexes involved in transcriptional regulation (including NELF and Integrator) so the mechanisms of promoter escape, pause-release and transcriptional elongation may be different. We have rephrased some of the conclusions of the ChIP-qPCR analysis more carefully (l. 273-285).

2. Are the differences in PolII-associated nascent transcripts between WT and cdk8 mutant significant (Fig. EV6)? These data look quite variable and I didn't see any statistical analysis in that figure. I think this is reasonably important in terms of mechanism.

>> We have added additional biological replicates and performed statistical analysis that fully supports our previous conclusions (Fig EV5B-C, l. 315-322). In addition, we have complemented the data with direct qRT-PCR analysis of unspliced transcripts (without NRPB1-pulldown). These fully support our conclusions and validate the technical approach (added as Fig EV5D, l. 322-327).

Minor points:

1. Are the binding sites/motifs for HSFA2 known, and if so - can these be shown on the ChIP-qPCR schematics relative to the primers used for the analysis. Based on the pAPX2::LUC reporter used - I presumed the HSFA2 binding sites are in the promoter fragment, but this was not clear from the text. This is important given the observations about Mediator/Cdk8 binding at the 3' end of the gene targets analyzed.

>> APX2, MIPS2, HSP70 and HSP101 all contain an HSFA2 binding site in their promoter. We have added the binding sites for HSFA2 to the schematics (as confirmed in Kappel et al., 2023) in Fig 5, Fig 7, Fig 8, Fig EV4.

2. The Abcam H3M4me3 antibody used for ChIP-qPCR is unfortunately not particularly specific for me3 vs me2 (see <https://doi.org/10.1016/j.molcel.2018.08.015>). I don't think this is problematic for the data shown here, but the authors should probably interpret their findings as reflecting some combination of H3K4me2 and me3 enrichment.

>> We thank the reviewer for this comment and have added this to the discussion (l. 355-356).

3. Most of the qPCR data (ChIP and RT) are shown with error bars as SEM, but I think SDEV would be more appropriate given the sample sizes. In addition, the authors should consider whether t-tests are appropriate for every comparison given that multiple genotypes are being compared. This is a relatively minor point, but some of these data is quite variable (as is typical for some of these experiments) and a more stringent criteria for identifying significant differences would help to focus on those that are most biologically relevant. A very minor point is that it's difficult to see the data points and error bars when bars are colored black, so selecting a different color for the wild-type controls would be helpful.

>> We thank the reviewer for these suggestions. The standard error of the mean (SEM) gives the confidence interval in which the true average is likely located, while standard deviation (SD) informs about the range of data variation. We consider SEM to be more information here, given that individual data points are plotted in all Figures, from which the range of experimental variation can be assessed.

Similarly, we consider t-test appropriate, because in many of the biochemical experiments we are interested to compare two genotypes. We agree that in many of the gene expression experiments we use multiple comparisons, so we have re-calculated the statistical analyses using first ANOVA and then Tukey's HSD, and plotted the results using compact letter display. In all cases, the conclusions were not affected by the new statistical analyses.

We have recoloured black bars throughout to allow the data points and error bars to be visible.

Optional point/question for authors:

4. Is there a super-elongation complex that functions in plants much like it does in mammalian cells? Although you discuss pausing/release as a potential mechanism in the discussion, I was unsure if Pol

II pausing/release work in the same way in plants as it does in animals. A brief outline of what's known in plants about these processes could be helpful to readers who are working in other systems.

>> *As far as we know, no complex similar to the SEC has been identified in plants, although most general elongation factors have been co-purified from elongating Pol II (Antosz et al (2017) Plant Cell 10.1105/tpc.16.00735). Polymerase pausing works differently in plants as there is no NELF or Integrator, yet there are nucleosome-defined mechanisms that determine mRNA synthesis efficiency. We have added the following sentence to the discussion (l. 404-406): "However, plant genomes lack homologues for a number of protein complexes involved in transcriptional regulation (including NELF and Integrator) so the mechanism of promoter escape, pausing and re-initiation may be quite different (Obermeyer et al, 2023)".*

Referee #3:

Crawford et al performed a genetic screen to identify regulators of heat-stress induced transcriptional memory (i.e. priming) in the model plant *Arabidopsis thaliana*. They identified the genes MED12 and CDK8, which are components of the dynamic module (CKM) of mediator transcriptional co-activator complex. Based in these findings, they carry out a comprehensive molecular and biochemical characterization of the role of CDK8 and MED12 in HS transcriptional memory. They demonstrate that HS induced the recruitment of CDK8 to target genes by the transcription factor HSFA2, and confirmed direct interaction between CDK8 and HSFA2 *in vivo*. They further show that CDK8 interacts with cMed, and is required for HS-induced H3K9me3 at the HS memory gene APX2, but surprisingly not at MIPS2 gene (another HS-memory gene). Last, they found that repeated HS leads to the binding of CDK8 along gene body of memory genes, potentially modulating the PolIII transcriptional activity. Intriguingly, in the absence of CDK8, non-productive PolIII accumulate at HS memory genes.

This is certainly an interesting work reporting novel finding that shed light on the role of CKM in the establishment of active chromatin states required for short-term transcriptional memory in response to HS. The manuscript is well written and results are presented with clarity and sufficient details.

I have nonetheless two concerns:

Lines 263-2. It is surprising that despite CDK8 binds to APX2 and MIPS2 gene locus, and is required for their transcriptional memory, *cdk8* mutant only affects H3K9me3 at APX2 (Figure 7). Indeed, MIPS2 has a very slightly decrease at only one probe, which given the large number of non-independent statistical tests performed can be by pure chance. The seemingly gene-specific function of CDK8 in H3K4me3 deposition at HS-memory genes weaken the statement "CDK8 is required for H3K9me3 accumulation at HS memory gene loci". The authors might want to consider to perform ChIP-seq experiment to evaluate the set of genes with varying H3K9me3 levels in wt and *cdk8* mutants in response to P+T. A

>> We thank the reviewer for these suggestions. In previous work we found that APX2 displays a stronger enrichment of H3K4me3 compared to other type II genes, such as MIPS2. Hence we expect APX2 to show the strongest effect on H3K4me3. We have clarified this in the text (l. 273-285) and quantified the statement in the discussion (l. 353).

Abstract: "CDK8 also binds to the 3' region of target genes, where it promotes elongation, termination or rapid initialisation of Pol II complex" It is unclear from the presented results whether specific binding to 3' region has any role on any of the activities mentioned here. Indeed, 3' binding is accompanied with gene body and promoter binding, making it impossible to assign a role to any

specific genic region. Also, Figure 8 clearly shows that the lack of CDK8 increases the occupancy of total as well as elongating PolII, contradicting the conclusion "promotes elongation, termination or rapid initialisation of Pol II complex"

These statements in the abstract should be qualified, or else supported by direct evidences showing a role for CDK8 on promoting elongation, termination or rapid initialisation, such as provided in

>> We have qualified the statement in the abstract as suggested. "In addition to the promoter and transcriptional start region, CDK8 also binds the 3'-region of target genes, where it may promote elongation, termination or rapid re-initiation of Pol II complexes during transcriptional memory bursts."

Dear Isabel,

We have now received re-review reports from two of the three referees that initially appraised your manuscript. As you will see, you have addressed their concerns satisfactorily. Before I can finally accept the manuscript though, I have noticed some small outstanding editorial points that will need to be addressed. In this regard would you please:

rename the Conflict of Interest statement the 'Disclosure and Competing Interests Statement',
remove the author credit section from the manuscript,
upload EV legends as a separate sheet in each Excel file,
ensure all datasets in public databases are made public upon the manuscript's acceptance,
provide a separate 'Data Information' section in the legends of figure EV 5b-d,
indicate the statistical test used for data analysis in the legends of figures 3b, 6e, EV 3b, and EV 5b-d,
correct a mismatch between the annotated p values in the figure legend and the figure file in figures 2e; 7; 8a-c; EV 1c-d,
define whiskers in the legend of figure 4d,
define n in the legends of figures 2f-g, 4d, EV 1c-d, EV 2c-d, EV 4 and EV 5d, and
define error bars in the legends of figures 2f-g, EV 1c-d, EV 2c-d and EV 5d.
In addition, figure legends should be placed after the References.
Please also provide me with a two-sentence general summary statement and 3-5 bullet points that capture the key findings of the paper.

We also need a summary figure for the synopsis. The size should be 550 wide by [200-400] high (pixels). You can also use something from the figures if that is easier.

Best wishes,

William

William Teale, PhD
Editor
The EMBO Journal
w.teale@embojournal.org

We realize that it is difficult to revise to a specific deadline. In the interest of protecting the conceptual advance provided by the work, we recommend a revision within 3 months (6th Mar 2024). Please discuss the revision progress ahead of this time with the editor if you require more time to complete the revisions. Use the link below to submit your revision:

Referee #2:

The authors have addressed all of my concerns from the previous review.

Referee #3:

The authors addressed all my concerns in the revised version of this manuscript.

Dear Isabel,

We have now received re-review reports from two of the three referees that initially appraised your manuscript. As you will see, you have addressed their concerns satisfactorily. Before I can finally accept the manuscript though, I have noticed some small outstanding editorial points that will need to be addressed. In this regard would you please:

rename the Conflict of Interest statement the 'Disclosure and Competing Interests Statement',
remove the author credit section from the manuscript,
upload EV legends as a separate sheet in each Excel file,
ensure all datasets in public databases are made public upon the manuscript's acceptance,
provide a separate 'Data Information' section in the legends of figure EV 5b-d,
indicate the statistical test used for data analysis in the legends of figures 3b, 6e, EV 3b, and EV 5b-d,
correct a mismatch between the annotated p values in the figure legend and the figure file in figures 2e; 7; 8a-c; EV 1c-d,
define whiskers in the legend of figure 4d,
define n in the legends of figures 2f-g, 4d, EV 1c-d, EV 2c-d, EV 4 and EV 5d, and
define error bars in the legends of figures 2f-g, EV 1c-d, EV 2c-d and EV 5d.
In addition, figure legends should be placed after the References.
Please also provide me with a two-sentence general summary statement and 3-5 bullet points that capture the key findings of the paper.

We also need a summary figure for the synopsis. The size should be 550 wide by [200-400] high (pixels). You can also use something from the figures if that is easier.

Best wishes,

William

Response:

Thank you very much for the positive reply and the interest in publishing our revised manuscript. We thank you for the editorial suggestions. We have added the requested information and made the requested editorial changes in the manuscript. The RNA-seq data are now publicly accessible at NCBI GEO (GSE232094).

Dear Isabel,

I am pleased to inform you that your manuscript has been accepted for publication in the EMBO Journal.

Congratulations on a great study; I'm sure many people will find it very interesting.

Best wishes,

William

William Teale, PhD
Editor
The EMBO Journal
w.teale@embojournal.org
